# New Bounds for Hyperparameter Tuning of Regression Problems Across Instances

**Maria-Florina Balcan**
Carnegie Mellon University
`ninamf@cs.cmu.edu`

**Anh Tuan Nguyen**
Carnegie Mellon University
`atnguyen@cs.cmu.edu`

**Dravyansh Sharma**
Carnegie Mellon University
`dravyans@cs.cmu.edu`

## Abstract

The task of tuning regularization coefficients in regularized regression models with provable guarantees across problem instances still poses a significant challenge in the literature. This paper investigates the sample complexity of tuning regularization parameters in linear and logistic regressions under $\ell_1$ and $\ell_2$-constraints in the data-driven setting. For the linear regression problem, by more carefully exploiting the structure of the dual function class, we provide a new upper bound for the pseudo-dimension of the validation loss function class, which significantly improves the best-known results on the problem. Remarkably, we also instantiate the first matching lower bound, proving our results are tight. For tuning the regularization parameters of logistic regression, we introduce a new approach to studying the learning guarantee via an approximation of the validation loss function class. We examine the pseudo-dimension of the approximation class and construct a uniform error bound between the validation loss function class and its approximation, which allows us to instantiate the first learning guarantee for the problem of tuning logistic regression regularization coefficients.

## 1 Introduction

Regularized linear models, including the Elastic Net [1], and Regularized Logistic Regression [2, 3, 4], as well as their variants [5, 6, 7], have found widespread use in diverse fields and numerous application domains. Thanks to their simplicity and interpretability, those methods are popular choices for controlling model complexity, improving robustness, and preventing overfitting by selecting relevant features [4, 8, 9]. Moreover, regularized linear models can be adapted to the non-linear regime using kernel methods [10, 11], significantly expanding their applicability to a wide range of problems. In typical applications, one needs to solve not only a single regression problem instance, but several related problems from the same domain. Can we learn how to regularize with good generalization across the related problem instances?

Suppose we have a regression dataset $(X, y) \in \mathbb{R}^{m \times p} \times \mathcal{Y}^m$, where $X$ is a design matrix with $m$ samples and $p$ features, and $y$ is a target vector. Regularized linear models aim to compute an estimator $\hat{\beta}^{(X,y)}(\lambda)$ by solving the optimization problem

$$\hat{\beta}^{(X,y)}(\lambda) = \underset{\beta \in \mathbb{R}^p}{\operatorname{argmin}} \left[ l(\beta, (X, y)) + \lambda_1 \|\beta\|_1 + \lambda_2 \|\beta\|_2^2 \right], \tag{1}$$

where $(\lambda_1, \lambda_2) \in \mathbb{R}_{\geq 0}^2$ are the regularization coefficients. For instance, if $\lambda \in \mathbb{R}_{>0}^2$, $y \in \mathbb{R}^m$, and $l(\beta, (X, y)) = \frac{1}{2}\|y - X\beta\|_2^2$ (squared-loss function), we get the well-known Elastic Net [1]. On the other hand, if $\lambda \in \{(\lambda_1, 0), (0, \lambda_2)\}$ for $\lambda_1, \lambda_2 > 0$, $y \in \{\pm 1\}^m$, and $l(\beta, (X, y)) = \frac{1}{m} \sum_{i=1}^m \log(1 + \exp(-y_i x_i^\top \beta))$, we obtain regularized logistic regression.

---

Correspondence: `atnguyen@cs.cmu.edu`

37th Conference on Neural Information Processing Systems (NeurIPS 2023).

In regularized linear models, the parameters $\lambda$ play a crucial role in controlling the sparsity ($\ell_1$) and shrinkage ($\ell_2$) constraints, and are essential in ensuring better generalization and robustness [9, 4, 12]. A popular approach in practice is cross-validation, which involves choosing a finite grid of values of $\lambda$ and iteratively solving the regression problem for multiple values of $\lambda$ and evaluating on held-out validation sets to determine the optimal parameter. Principled techniques with theoretical guarantees suffer from various limitations, for example require strong assumptions about the original problem [13], or aim to search the optimal parameter over a discrete subset instead of the whole continuous domain. Moreover, repeatedly solving the regression problem is particularly inefficient if we have multiple problem instances from the same problem domain.

In this work, we investigate an alternative setting for tuning regularization parameters, namely *data-driven algorithm design*, following the previous line of work by Balcan et al. [14]. Unlike the traditional approach, which involves considering a single dataset $(X, y)$, in the data-driven approach, we analyze a collection of datasets or problem instances $(X^{(i)}, y^{(i)}, X_{\text{val}}^{(i)}, y_{\text{val}}^{(i)})$ drawn from an underlying problem distribution $\mathcal{D}$. Our objective is to determine the optimal regularization parameters $\lambda$ so that when using the training set $(X^{(i)}, y^{(i)})$ and $\lambda$ to select a model in Optimization problem 1, the selected model minimizes loss on the validation set $(X_{\text{val}}^{(i)}, y_{\text{val}}^{(i)})$. As remarked by Balcan et al. [14], data-driven algorithm design can handle more diverse data generation scenarios in practice, including cross validation and multitask-learning [15, 16]. We emphasize that the data-driven setting differs significantly from the standard single dataset setting.

In this paper, we consider the problem of tuning regularization parameters in regularized logistic regression and the Elastic Net across multiple problem instances. Our contributions are:

- We present an improved upper bound (Theorem 3.3) on the pseudo-dimension for tuning the Elastic Net regularization parameters across problem instances by establishing a novel structural result for the validation loss function class (Theorem 3.2). We provide a crucial refinement to the piecewise structure of this function class established by Balcan et al. [14], by providing a bound on the number of distinct functional behaviors across the pieces. This enables us to describe the computation of the validation loss function as a GJ algorithm [17], which yields an upper-bound of $O(p)$ on the pseudo-dimension, a significant improvement of the prior best bound of $O(p^2)$ by Balcan et al. [14], and a corresponding improvement in the sample complexity (Theorem 3.4).

- Furthermore, we establish the tightness of our result by providing the first asymptotically matching lower bound of $\Omega(p)$ on the pseudo-dimension (Theorem 3.5). It is worth noting that our results have direct implications for other specialized cases, such as LASSO and Ridge Regression.

- We further extend our results on the Elastic Net to regularized kernel linear regression problem (Corollary 3.6).

- We propose a novel approach to analyze the problem of tuning regularization parameters in regularized logistic regression, which involves indirectly investigating an approximation of the validation loss function class. Using this approach, we instantiate the first learning guarantee for this problem in the data-driven setting (Theorem 4.4).

## 1.1 Related work

**Model selection for regularized linear models.** Extensive research has focused on the selection of optimal parameters for regularized linear models, including the Elastic Net and regularized logistic regression. This process usually entails choosing the appropriate regularization coefficients for a given dataset [18, 19]. Nevertheless, a substantial proportion of this research relies on heuristic approaches that lack theoretical guarantees [20, 21]. Others have concentrated on creating tuning objectives that go beyond validation error [22, 23], but with no clearly defined procedures for provably optimizing them. The conventional method for selecting a tuning regularization parameter is through grid-based selection, which aims to choose the parameter from a subset, known as a grid, within the parameter space. While this approach provides certain guarantees [24], it falls short in delivering an optimal solution across the entire continuous parameter space, particularly when using tuning objectives that exhibit numerous discontinuities. Additionally, the grid-based technique is highly sensitive to density, as selecting a grid that is either too dense or too coarse might result in inefficient search or highly inaccurate solutions. Other guarantees require strong assumptions on the data distribution, such as sub-Gaussian noise [25, 13]. Some studies focus on evaluating regularized linear models by

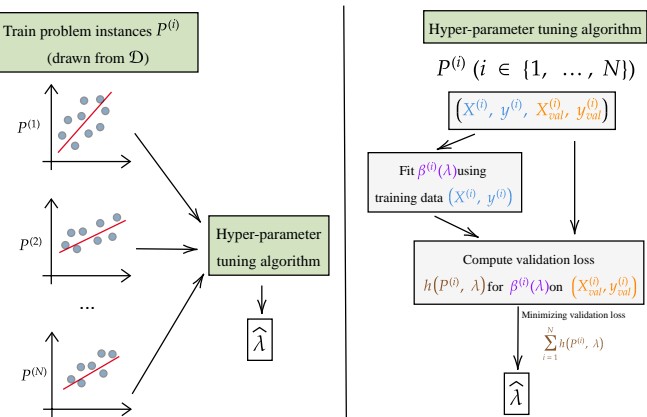

Figure 1: The process of tuning regularization parameter $\lambda$ across problem instances. Given a set of $N$ problem instances $\{P^{(1)}, \ldots, P^{(N)}\}$ drawn from some problem distribution $\mathcal{D}$, one seeks to choose the best parameter $\hat{\lambda}$ by minimizing the total validation loss $\sum_{i=1}^{N} h(P^{(i)}; \lambda)$.

constructing solution paths [26, 2, 27]. However, it is important to note that these approaches are primarily computational in nature and do not provide theoretical guarantees.

**Data-driven algorithm design.** Data-driven algorithms can adapt their internal structure or parameters to problem instances from unknown application-specific distributions. It is proved to be effective for a variety of combinatorial problems, such as clustering, integer programming, auction design, and graph-based semi-supervised learning [28, 29, 30, 31]. Balcan et al. [14] recently introduced a novel approach to tuning regularization parameters in regularized linear regression models, such as Elastic Net and its variants. They applied data-driven analysis to reveal the underlying discrete structure of the problem and leveraged a general result from [32] to obtain an upper bound on the pseudo-dimension of the problem. To provably tune the regularization parameters across problem instances, they proposed a simple ERM learner and provided sample complexity guarantee for such learner. However, the general techniques from [32] do not always lead to optimal bounds on the pseudodimension. Our paper is an example of a problem where these bounds (as derived in [14]) are sub-optimal, and more specialized techniques due to [31] result in the tighter bounds that we obtain. Also prior work does not establish any lower bound on the pseudodimension. Furthermore, it should be noted that their analysis heavily relies on the assumption of having a closed-form representation of the Elastic Net estimator [27]. This approach may not be applicable in analyzing other regularized linear models, such as regularized logistic regression, for which we propose an alternative approach.

## 2 Problem setting

In this section, we provide a formal definition of the problem of tuning regularization parameters in the Elastic Net and regularized logistic regression (RLR) across multiple problem instances, which follows the settings by Balcan et al. [14]. Given a problem instance $P = (X, y, X_{\text{val}}, y_{\text{val}})$, where $(X, y) \in \mathbb{R}^{m \times p} \times \mathcal{Y}^m$ represents the training dataset with $m$ samples and $p$ features, and $(X_{\text{val}}, y_{\text{val}}) \in \mathbb{R}^{m' \times p} \times \mathcal{Y}^{m'}$ denotes the validation split with $m'$ samples, we consider the estimator $\hat{\beta}_{(X,y)}(\lambda)$ defined as:

$$\hat{\beta}_{(X,y)}(\lambda) \in \underset{\beta \in \mathbb{R}^p}{\operatorname{argmin}} \, l(\beta, (X, y)) + \langle \lambda, R(\beta) \rangle, \tag{2}$$

where $l(\beta, (X, y))$ represents the objective loss function, $\lambda$ denotes the regularization coefficients, and $R(\beta) = (\|\beta\|_1, \|\beta\|_2^2)$ represents the regularization vector function.

For instance, if $\lambda \in \mathbb{R}_{>0}^2$, $\mathcal{Y} \equiv \mathbb{R}$, and $l(\beta, (X, y)) = l_{\text{EN}}(\beta, (X, y)) = \frac{1}{2m} \|y - X\beta\|_2^2$, we get the well-known Elastic Net. On the other hand, if $\lambda \in \{(\lambda_1, 0), (0, \lambda_2)\}$ for $\lambda_1, \lambda_2 > 0$, $y \in \{\pm 1\}^m$, and $l(\beta, (X, y)) = l_{\text{RLR}}(\beta, (X, y)) = \frac{1}{m} \sum_{i=1}^{m} \log(1 + \exp(-y_i x_i^\top \beta))$, we obtain RLR with $\ell_1$ or $\ell_2$ regularization. Note that for the Elastic Net hyperparameter tuning problem, we allows the regularization coefficients of both $\ell_1, \ell_2$ are positive, while in the Regularized Logistic Regression

problem, we consider either $\ell_1$ or $\ell_2$ as the regularization term. We then use the validation set $(X_{\text{val}}, y_{\text{val}})$ to calculate the validation loss $h(\lambda, P) = l(\hat{\beta}_{(X,y)}(\lambda), (X_{\text{val}}, y_{\text{val}}))$ corresponding to the problem instance $P$ and learned regularization parameters $\lambda$.

In the *data-driven setting*, we receive a collection of $n$ problem instances $P^{(i)} = (X^{(i)}, y^{(i)}, X_{\text{val}}^{(i)}, y_{\text{val}}^{(i)}) \in \mathcal{R}_{m_i, p_i, m_i'}$ for $i \in [n]$, where $\mathcal{R}_{m_i, p_i, m_i'} := \mathbb{R}^{m_i \times p_i} \times \mathcal{Y}^{m_i} \times \mathbb{R}^{m_i' \times p_i} \times \mathcal{Y}^{m_i'}$. The problem space $\Pi_{m,p}$ is given by $\Pi_{m,p} = \cup_{m_1 \geq 0, m_2 \leq m, p_1 \leq p} \mathcal{R}_{m_1, p_1, m_2}$, and we assume that problem instance $P$ is drawn i.i.d from the problem distribution $\mathcal{D}$ over $\Pi_{m,p}$. Remarkably, in this setting, problem instances can have varying training and validation sample sizes, as well as different sets of features. This general framework applies to practical scenarios where the feature sets differ among instances and allows one to learn regularization parameters that effectively work on average across multiple different but related problem instances. See Figure 1 for an illustration of the setting.

The goal here is to learn the value $\hat{\lambda}$ s.t. with high probability over the draw of $n$ problem instances, the expected validation loss $\mathbb{E}_{P \sim \mathcal{D}} h(\hat{\lambda}, P)$ is close to $\min_\lambda \mathbb{E}_{P \sim \mathcal{D}}[h(\lambda, P)]$. This paper primarily focuses on providing learning guarantees in terms of sample complexity for the problem of tuning regularization parameters in the Elastic Net and regularized logistic regression (RLR). Specifically, we aim to address the question of how many problem instances are required to learn a value of $\lambda$ that performs well across all problems $P$ drawn from the problem distribution $\mathcal{D}$. To achieve this, we analyze the pseudo-dimension (in the case of the Elastic Net) or the Rademacher Complexity (for RLR) of the validation loss function class $\mathcal{H} = \{h(\lambda, \cdot) \mid \lambda \in \Lambda\}$, where $\Lambda$ represents the search space for $\lambda$.

# 3 Tight pseudo-dimension bounds for Elastic Net hyperparameter tuning

In this section, we will present our results on the pseudo-dimension upper and lower bounds for the regularized linear regression problem in the data-driven setting. Classic learning-theoretic results [33, 34] connect the pseudo-dimension of the validation loss function class (parameterized by the regularization coefficient) with the *sample complexity* of the number of problem instances $\{P^{(1)}, \ldots, P^{(n)}\}$ drawn i.i.d. from some unknown problem distribution $\mathcal{D}$ needed for learning good regularization parameters with high confidence. Let $h_{\text{EN}}(\lambda, P) = l_{\text{EN}}(\hat{\beta}_{(X,y)}(\lambda), (X_{\text{val}}, y_{\text{val}}))$ be the validation loss function of the Elastic Net, and $\mathcal{H}_{\text{EN}} = \{h_{\text{EN}}(\lambda, P) : \Pi_{m,p} \to \mathbb{R}_{\geq 0} \mid \lambda \in \mathbb{R}_{>0}^2\}$ be the corresponding validation loss function class, we now present tight bounds for $\text{Pdim}(\mathcal{H}_{\text{EN}})$.

## 3.1 The Goldberg-Jerrum framework

Recently, Bartlett et al. [31] instantiate a simplified version of the well-known Goldberg-Jerrum (GJ) Framework [17]. The GJ framework offers a general pseudo-dimension upperbound for a wide class of functions in which each function can be computed by a *GJ algorithm*. We provide a brief overview of the GJ Framework which is useful in establishing our improved pseudo-dimension upper bound.

**Definition 1** (GJ Algorithm, [31]). *A **GJ algorithm** $\Gamma$ operates on real-valued inputs, and can perform two types of operations:*

- *Arithmetic operators of the form $v'' = v \odot v'$, where $\odot \in \{+, -, \times, \div\}$, and*

- *Conditional statements of the form "if $v \geq 0 \ldots$ else $\ldots$".*

*In both cases, $v$ and $v'$ are either inputs or values previously computed by the algorithm.*

General speaking, each intermediate value of the GJ algorithm $\Gamma$ can be described by a *rational function*, which is a fractional between two polynomials, of the algorithm's inputs. The degree of a rational function is equal to the maximum degree of the polynomials in its numerator and its denominator. We can define two quantities that represent the complexity of GJ algorithms.

**Definition 2** (Complexity of GJ algorithm, [31]). *The **degree** of a GJ algorithm is the maximum degree of any rational function it computes of the inputs. The **predicate complexity** of a GJ algorithm is the number of distinct rational functions that appear in its conditional statements.*

The following theorem essentially shows that for any function class $\mathcal{F}$, if we can describe any function $f \in \mathcal{F}$ by a GJ algorithm of which the degree and predicate complexity are at most $\Delta$ and $\Lambda$, respectively, then we can automatically obtain the upper bound for the pseudo-dimension of $\mathcal{F}$.

**Theorem 3.1** ([31]). *Suppose that each function $f \in \mathcal{F}$ is specified by $n$ real parameters. Suppose that for every $x \in \mathcal{X}$ and $r \in \mathbb{R}$, there is a GJ algorithm $\Gamma_{x,r}$ that given $f \in \mathcal{F}$, returns "true" if $f(x) \geq r$ and "false" otherwise. Assume that $\Gamma_{x,r}$ has degree $\Delta$ and predicate complexity $\Lambda$. Then, $\mathrm{Pdim}(\mathcal{F}) = O(n \log(\Delta\Lambda))$.*

## 3.2 Upper bound

Our work improves on prior research [14] by presenting an upper bound on the pseudo-dimension of Elastic Net validation loss function class $\mathcal{H}_{\mathrm{EN}}$ parameterized by $\lambda$. We extend the previous piecewise-decomposable structure of the loss function by providing a bound on the number of distinct rational piece functions for any fixed problem instance (Definition 3). This allows us to use a GJ algorithm and Theorem 3.1 to obtain better bounds on the number of distinct predicates that need to be computed. While prior research only used a bound on the number of distinct loss function pieces generated by algebraic boundaries, our new observation that the loss function has a limited number of possible distinct functional behaviors yields a tighter upper bound on the pseudo-dimension (Theorem 3.2). In Theorem 3.5, we will demonstrate the tightness of our upper bound by providing a novel lower bound for the problem.

We first provide a refinement of the piece-wise decomposable function class terminology introduced by [32] which is useful for establishing our improved upper bound. Intuitively, this corresponds to real-valued functions for which the domain is partitioned by finitely many *boundary functions* such that the function is well-behaved in each piece in the partition, i.e. can be computed using a *piece function* from another function class.

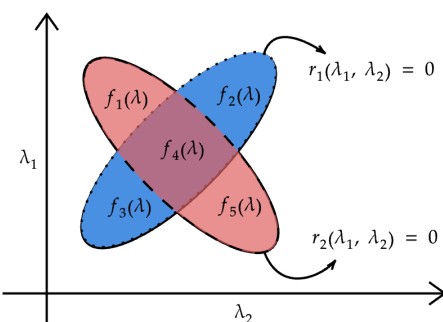

Figure 2: An illustration of piece-wise structure of $\mathcal{H}_{EN}^* = \{h_P^* : \mathcal{H}_{EN} \to \mathbb{R}_{\geq 0} \mid P \in \Pi_{m,p}\}$. Given a problem instance $P$, the function $h_P^*(\lambda) = h(P; \lambda)$ is a fixed rational function $f_i(\lambda)$ in each piece (piece function), that is regulated by boundary functions $g_{r_i}$ of the form $\mathbb{1}\{r_i(\lambda) < 0\}$. As mentioned in our main result, there are at most $3^p$ functions $f_i$ of degree at most $2p$, and at most $p3^p$ functions $g_{r_i}$ where $r_i$ is a polynomial of degree at most $p$.

**Definition 3.** *A function class $\mathcal{H} \subseteq \mathbb{R}^{\mathcal{Y}}$ that maps a domain $\mathcal{Y}$ to $\mathbb{R}$ is $(\mathcal{F}, k_{\mathcal{F}}, \mathcal{G}, k_{\mathcal{G}})-$piece-wise decomposable for a class $\mathcal{G}$ of boundary functions and a class $\mathcal{F} \in \mathbb{R}^{\mathcal{Y}}$ of piece functions if the following holds: for every $h \in \mathcal{H}$, (1) there are $k_{\mathcal{G}}$ functions $g^{(1)}, \ldots, g^{(k_{\mathcal{G}})} \in \mathcal{G}$ and a function $f_{\boldsymbol{b}} \in \mathcal{F}$ for each bit vector $\boldsymbol{b} \in \{0,1\}^{k_{\mathcal{G}}}$ s.t. for all $y \in \mathcal{Y}$, $h(y) = h_{\boldsymbol{b}_y}(y)$ where $\boldsymbol{b}_y = \{(g^{(1)}(y), \ldots, g^{(k_{\mathcal{G}})}(y))\} \in \{0,1\}^{k_{\mathcal{G}}}$, and (2) there is at most $k_{\mathcal{F}}$ different functions in $\mathcal{F}$.*

A key distinction from [32] is the finite bound $k_{\mathcal{F}}$ on the number of different piece functions needed to define any function in the class $\mathcal{H}$. Under this definition we give the following more refined structure for the Elastic Net loss function class by extending arguments from [14].

**Theorem 3.2.** *Let $\mathcal{H}_{EN} = \{h_{EN}(\lambda, \cdot) : \Pi_{m,p} \to \mathbb{R}_{\geq 0} \mid \lambda \in \mathbb{R}_{>0}^2\}$ be the class of Elastic Net validation loss function class. Consider the dual class $\mathcal{H}_{EN}^* = \{h_P^* : \mathcal{H}_{EN} \to \mathbb{R}_{\geq 0} \mid P \in \Pi_{m,p}\}$, where $h_P^*(h_{EN}(\lambda, \cdot)) = h_{EN}(\lambda, P)$. Then $\mathcal{H}_{EN}^*$ is $(\mathcal{F}, 3^p, \mathcal{G}, p3^p)$-piecewise decomposable, where the piece function class $\mathcal{F} = \{f_q : \mathcal{H}_{EN} \to \mathbb{R}\}$ consists at most $3^p$ rational functions $f_{q_1,q_2} : h_{EN}(\lambda, \cdot) \mapsto \frac{q_1(\lambda_1,\lambda_2)}{q_2(\lambda_1,\lambda_2)}$ of degree at most $2p$, and the boundary function class $\mathcal{G} = \{g_r : \mathcal{H}_{EN} \to \{0,1\}\}$ consists*

*of semi-algebraic sets bounded by at most $p3^p$ algebraic curves $g_r : h_{EN}(\lambda, \cdot) \mapsto \mathbb{1}\{r(\lambda_1, \lambda_2) < 0\}$, where $r$ is a polynomial of degree at most $p$.*

Figure 2 demonstrates the piece-wise structure of $\mathcal{H}^*_{EN}$, which allows us to establish an improved upper bound on the pseudo-dimension.

**Theorem 3.3.** *Let $\mathcal{H}_{EN} = \{h_{EN}(\lambda, \cdot) : \Pi \to \mathbb{R}_{\geq 0} \mid \lambda \in \mathbb{R}^2_{>0}\}$ be the Elastic Net validation loss function class that maps problem instance $P$ to validation loss $h_{val}(\lambda, P)$. Then $\mathrm{Pdim}(\mathcal{H}_{EN})$ is $O(p)$.*

*Proof Sketch.* For every problem instance $P \in \Pi_{m,p}$, and a threshold $r \in \mathbb{R}$, consider the computation $\mathbb{1}\{h_{EN}(\lambda, P) - r \geq 0\}$ for any $h_{EN}(\lambda, \cdot) \in \mathcal{H}_{EN}$. From Theorem 3.2, we can describe $\mathbb{1}\{h_{EN}(\lambda, P) - r \geq 0\}$ as a GJ algorithm $\Gamma_{P,r}$ which is specified by 2 parameters $\lambda_1, \lambda_2$, has degree of at most $2p$, and has predicate complexity of at most $(p+1)3^p$ (See Figure 3). Then Theorem 3.1 implies that $\mathrm{Pdim}(\mathcal{H}_{EN}) = O(p)$. $\qquad\square$

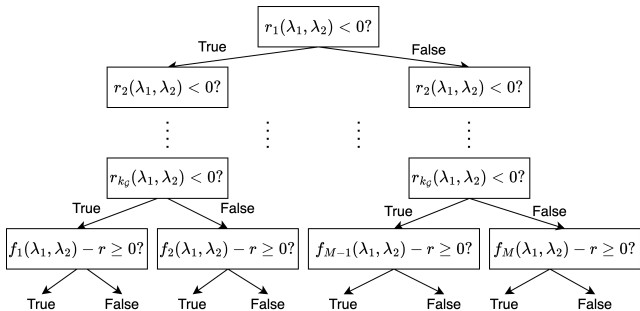

Figure 3: An illustration of how $\mathbb{1}\{h_{EN}(\lambda, P) - r \geq 0\}$ is computed as a GJ algorithm. The number of boundary (polynomial) functions $k_{\mathcal{G}}$ is at most $p3^p$, and there are at most $M = 3^p$ distinct (rational) piece functions. All the polynomial and rational functions are of degree at most $2p$.

The detailed proof of Theorem 3.3 can be found on Appendix B.1.2. Recent work by Balcan et al. [14] also studied the Elastic Net, and showed the piece-wise structure of the dual function of the validation loss function which implies an upper bound of $O(p^2)$ by employing the general tool from [32]. We establish a tighter bound of $O(p)$ in Theorem 3.3 by establishing additional properties of the loss function class and giving a GJ algorithm for computing the loss functions.

To guarantee the boundedness of the considered validation loss function classes, we will have the following assumptions for the data and regularization parameters. The first assumption is that all features and target values in the training and validation examples are bounded. The second assumption is that we only consider regularization coefficient values $\lambda$ within an interval $[\lambda_{\min}, \lambda_{\max}]$. In practice, those assumptions are naturally satisfied by data normalization.

**Assumption 1** (Bounded covariate and label). *We assume that all the feature vectors and target values in training and validation set is upper-bounded by absolute constants $R_1$ and $R_2$, i.e. $\max\{\|X\|_\infty, \|X_{val}\|_\infty\} \leq R_1$, and $\max\{\|y\|_\infty, \|y_{val}\|_\infty\} \leq R_2$.*

**Assumption 2** (Bounded Coefficient). *We assume that $\lambda \in [\lambda_{\min}, \lambda_{\max}]^2$ with $\lambda_{\min} > 0$.*

Under Assumptions 2, 1, Theorem 3.3 immediately implies the following generalization guarantee for Elastic Net hyperparameter tuning.

**Theorem 3.4.** *Let $\mathcal{D}$ be an arbitrary distribution over the problem instance space $\Pi_{m,p}$. Under Assumptions 1, 2, the loss functions in $\mathcal{H}_{EN}$ have range bounded by some constant $H$ (Lemma C.1). Then there exists an algorithm s.t. for any $\epsilon, \delta > 0$, given $N = O(\frac{H^2}{\epsilon^2}(p + \log(\frac{1}{\delta})))$ sample problem instances drawn from $\mathcal{D}$, the algorithm outputs a regularization parameter $\hat{\lambda}$ such that with probability at least $1 - \delta$, $\mathbb{E}_{P \sim \mathcal{D}} h_{EN}(\hat{\lambda}, P) < \min_\lambda \mathbb{E}_{P \sim \mathcal{D}} h_{EN}(\lambda, P) + \epsilon$.*

*Proof.* Denote $\lambda^* = \arg\min_\lambda \mathbb{E}_{P \sim \mathcal{D}} h_{EN}(\lambda, P)$. From Theorems 3.3 and A.2, given $n = O(\frac{H^2}{\epsilon^2}(p + \log(\frac{1}{\delta})))$ problem instances $P^{(i)}$ for $i \in [N]$ drawn from $\mathcal{D}$, w.p. $1 - \delta$, we have $\mathbb{E}_{P \sim \mathcal{D}} h_{EN}(\hat{\lambda}, P) < \frac{1}{N}\sum_{i=1}^N h_{EN}(\hat{\lambda}, P^{(i)}) + \frac{\epsilon}{2} < \frac{1}{N}\sum_{i=1}^N h_{EN}(\lambda^*, P^{(i)}) + \frac{\epsilon}{2} < \mathbb{E}_{P \sim \mathcal{D}} h_{EN}(\lambda^*, P) + \epsilon$. $\qquad\square$

### 3.3 Lower bound

Remarkably, we are able to establish a matching lower bound on the pseudo-dimension of the Elastic Net loss function class, parameterized by the regularization parameters. Note that every Elastic Net problem can be converted to an equivalent LASSO problem [1]. In fact, we show something stronger, that the pseudo-dimension of even the LASSO regression loss function class (parameterized by regression coefficient $\lambda_1$) is $\Omega(p)$, from which the above observation follows (by taking $\lambda_2 = 0$ in our construction). Our proof of the lower bound adapts the "adversarial strategy" of [35] which is used to design a worst-case LASSO regularization path. While [35] construct a single dataset to bound the number of segments in the piecewise-linear LASSO solution path, we create a collection of problem instances for which all above-below sign patterns may be achieved by selecting regularization parameters from different segments of the solution path.

**Theorem 3.5.** *Let $\mathcal{H}_{LASSO}$ be a set of functions $\{h_{LASSO}(\lambda, \cdot) : \Pi_{m,p} \to \mathbb{R}_{\geq 0} \mid \lambda \in \mathbb{R}^+\}$ that map a regression problem instance $P \in \Pi_{m,p}$ to the validation loss $h_{LASSO}(\lambda, P)$ of LASSO trained with regularization parameter $\lambda$. Then $\mathrm{Pdim}(\mathcal{H}_{LASSO})$ is $\Omega(p)$.*

*Proof Sketch.* Consider $N = p$ problem instances for LASSO regression given by $P^{(i)} = (X^{(i)}, y^{(i)}, X_{\mathrm{val}}^{(i)}, y_{\mathrm{val}}^{(i)})$, where the training set $(X^{(i)}, y^{(i)}) = (X^*, y^*)$ is fixed and set using the "adversarial strategy" of [35], Proposition 2. The validation sets are given by single examples $(X_{\mathrm{val}}^{(i)}, y_{\mathrm{val}}^{(i)}) = (\mathbf{e}_i, 0)$, where $\mathbf{e}_i$ are standard basis vectors in $\mathbb{R}^p$. We will now proceed to provide the witnesses $r_1, \ldots, r_N$ and $\lambda$ values to exhibit a pseudo-shattering of these problem instances.

Corresponding to subset $T \subseteq [p]$ of problem instances, we will provide a value of $\lambda_T$ such that, we have $\ell_{\mathrm{LASSO}}(\lambda_T, P^{(i)}) > r_i$ iff $i \in T$, for each $i \in [p]$ and each $T \subseteq [p]$. We set all witnesses $r_i = 0$ for all $i \in [p]$. As a consequence of Theorem 1 in [35], the regularization path of $(X^*, y^*)$ consists of a linear segment corresponding all $2^p$ unsigned sparsity patterns in $\{0, 1\}^p$ (we will not need all the segments in the construction, but note that it is guaranteed to contain all distinct unsigned sparsity patterns) and we select $\lambda_T$ as any interior point corresponding to a linear segment with sparsity pattern $\{(c_1, \ldots, c_p) \mid c_i = 0 \text{ iff } i \in T\}$, i.e. elements in $T$ are exactly the ones with sparsity pattern 0. Therefore, $|\beta_T^* \cdot \mathbf{e}_i| = 0$ iff $i \in T$, where $\beta_T^*$ is the LASSO regression fit for regularization parameter $\lambda_T$. This implies the desired shattering condition w.r.t. witnesses $r_1 = 0, \ldots, r_N = 0$. Therefore, $\mathrm{Pdim}(\mathcal{H}_{\mathrm{LASSO}}) \geq p$. See Appendix B.2 for a full proof. $\qquad\square$

### 3.4 Hyperparameter tuning in Regularized Kernel Regression

The Kernel Least Squares Regression ([4]) is a natural generalization of the linear regression problem, which uses a kernel to handle non-linearity. In this problem, each sample has $p_1$ feature, corresponding to a real-valued target. Formally, each problem instance $P$ drawn from $\Pi$ can be described as

$$P = (X, y, X_{\mathrm{val}}, y_{\mathrm{val}}) \in \mathbb{R}^{m \times p_1} \times \mathbb{R}^m \times \mathbb{R}^{m' \times p_1} \times \mathbb{R}^{m'}.$$

A common issue in practice is that the relation between $y$ and $X$ is non-linear in the original space. To overcome this issue, we consider the mapping $\phi : \mathbb{R}^{p_1} \to \mathbb{R}^{p_2}$ which maps the original input space to a new feature space in which we hopefully can perform linear regression. Define $\phi(X) = (\phi(x_1), \ldots, \phi(x_m))_{m \times p_2}$, our goal is to find a vector $\theta \in \mathbb{R}^{p_2}$ so that the squared loss $\frac{1}{2} \|y - \phi(X)\theta\|_2^2 + R(\|\theta\|)$ is minimized, where the regularization term $R(\|\theta\|)$ is any strictly monotonically increasing function of the Hilbert space norm. It is well-known from the literature (e.g. [36]) that under the Representer Theorem's conditions, the optimal value $\theta^*$ can be linearly represented by row vectors of $\phi(X)$, i.e., $\theta^* = \phi(X)\beta = \sum_{i=1}^m \phi(x_i)\beta_i$, where $\beta = (\beta_1, \ldots, \beta_m) \in \mathbb{R}^m$. This directly includes the $\ell_2$ regularizer but does not include $\ell_1$ regularization. To overcome this issue, Roth ([3]) proposed an alternative approach to regularized kernel regression, which directly restricts the representation of coefficient $\theta$ via a linear combination of $\phi(x_i)$, for $i \in [m]$. The regularized kernel regression hence can be formulated as

$$\hat{\beta}_{l,\lambda}^{(X,y)} = \underset{\beta \in \mathbb{R}^m}{\arg\min} \frac{1}{2} \|y - K\beta\|_2^2 + \lambda_1 \|\beta\|_1 + \lambda_2 \|\beta\|_2^2,$$

where $k(x, x') = \langle \phi(x), \phi(x') \rangle$ is the kernel mapping, and the Gram matrix $K$ satisfies $[K]_{i,j} = k(x_i, x_j)$ for all $i, j \in [m]$.

Clearly, the problem above is a linear regression problem. Formally, denote $h_{\text{KER}}(\lambda, P) = \frac{1}{2}\|y - K\hat{\beta}_{(X,y)}(\lambda)\|_2$ and let $\mathcal{H}_{\text{KER}} = \{h_{\text{KER}}(\lambda, \cdot) : \Pi_{m,p} \to \mathbb{R}_{\geq 0} \mid \lambda \in \mathbb{R}_+^2\}$. The following result is a direct corollary of Theorem 3.3, which gives an upper bound for the pseudo-dimension of $\mathcal{H}_{\text{KER}}$.

**Corollary 3.6.** $\text{Pdim}(\mathcal{H}_{KER}) = O(m)$.

Note that $m$ here denotes the training set size for a single problem instance, and Corollary 3.6 implies a guarantee on the number of problem instances needed for learning a good regularization parameter for kernel regression via classic results [33, 34]. Our results do not make any assumptions on the $m$ samples within a problem instance/dataset; if these samples within problem instances are further assumed to be i.i.d. draws from some data distribution (distinct from problem distribution $\mathcal{D}$), then well-known results imply that $m = O(k \log p)$ samples are sufficient to learn the optimal LASSO coefficient [37, 38], where $k$ denotes the number of non-zero coefficients in the optimal regression fit.

## 4  Hyperparameter tuning for Regularized Logistic Regression

Logistic regression is more naturally suited to applications modeling probability of an event, like medical risk for a patient [39], predicting behavior in markets [40], failure probability of an engineering system [41] and many more applications [42]. It is a fundamental statistical technique for classification, and regularization is again crucial for avoiding overfitting and estimating variable importance. In this section, we will present learning guarantees for tuning the Regularized Logistic Regression (RLR) regularization coefficients across instances. Given a problem instance $P$ drawn from a problem distribution $\mathcal{D}$ over $\Pi_{m,p}$, let $h_{\text{RLR}}(\lambda, P) = l_{\text{RLR}}(\hat{\beta}_{(X,y)}(\lambda), (X_{\text{val}}, y_{\text{val}}))$ be the RLR validation loss function class (defined in Section 2), and let $\mathcal{H}_{\text{RLR}} = \{h_{\text{RLR}}(\lambda, \cdot) : \Pi_{m,p} \to \mathbb{R}_{\geq 0} \mid \lambda \in \mathbb{R}_{>0}\}$ be the RLR validation loss function class, our goal is to provide a learning guarantee for $\mathcal{H}_{\text{RLR}}$. Besides, we also study the commonly used 0-1 validation loss function class $\mathcal{H}_{\text{RLR}}^{0\text{-}1} = \{h_{\text{RLR}}^{0\text{-}1}(\lambda, \cdot) : \Pi_{m,p} \to \mathbb{R}_{\geq 0} \mid \lambda \in \mathbb{R}_{>0}\}$, where $h_{\text{RLR}}^{0\text{-}1}(\lambda, P) = \frac{1}{m'}\sum_{i=1}^{m'} \mathbb{1}\{y_i x_i^\top \hat{\beta}_{X,y}(\lambda) \leq 0\}$, which we will cover in Section 4.3. Similarly, to guarantee the boundedness of $\mathcal{H}_{\text{RLR}}$, we also assume that Assumptions 1 and 2 also hold in this setting.

### 4.1  Approximate solutions of Regularized Logistic Regression

The main challenge in analyzing the regularized logistic regression, unlike the regularized logistic regression problem, is that the solution $\hat{\beta}_{(X,y)}(\lambda)$ corresponding to a problem instance $P$ and particular value $\lambda > 0$ does not have a closed form depending on $\lambda$. We then propose an alternative approach to this end, which is examining via the approximation $\beta_{(X,y)}^{(\epsilon)}(\lambda)$ of the solution $\hat{\beta}_{(X,y)}(\lambda)$.

---

**Algorithm 1** Approximate incremental quadratic algorithm for RLR with $\ell_1$ penalty, [2]

---

Set $\beta_0^{(\epsilon)} = \hat{\beta}_{(X,y)}(\lambda_{\min})$, $t = 0$, small constant $\delta \in \mathbb{R}_{>0}$, and $\mathcal{A} = \{j \mid [\hat{\beta}_{(X,y)}(\lambda_{\min})]_j \neq 0\}$.
**while** $\lambda_t < \lambda_{\max}$ **do**

$\quad \lambda_{t+1} = \lambda_t + \epsilon$

$\quad \left(\beta_{t+1}^{(\epsilon)}\right)_{\mathcal{A}} = \left(\beta_t^{(\epsilon)}\right)_{\mathcal{A}} - \left[\nabla^2 l\left(\beta_t^{(\epsilon)}, (X,y)\right)_{\mathcal{A}}\right]^{-1} \cdot \left[\nabla l\left(\beta_t^{(\epsilon)}, (X,y)\right)_{\mathcal{A}} + \lambda_{t+1}\,\text{sgn}\left(\beta_t^{(\epsilon)}\right)_{\mathcal{A}}\right]$

$\quad \left(\beta_{t+1}^{(\epsilon)}\right)_{-\mathcal{A}} = \vec{0}$

$\quad \mathcal{A} = \mathcal{A} \cup \{j \neq \mathcal{A} \mid \nabla l(\beta_{t+1}^{(\epsilon)}, (X,y)) > \lambda_{t+1}\}$

$\quad \mathcal{A} = \mathcal{A} \setminus \{j \in \mathcal{A} \mid \left|\beta_{t+1,j}^{(\epsilon)}\right| < \delta\}$

$\quad t = t + 1$

---

The approximation Algorithm 1 (Algorithm 2) for the solution $\hat{\beta}(\lambda)$ of RLR under $\ell_1$ (or $\ell_2$) constraint were first proposed by Rosset [26, 2]. Given a problem instance $P$, and a sufficiently small step-size $\epsilon > 0$, using Algorithms 1, 2 yields an approximation $\beta_{(X,y)}^{(\epsilon)}$ of $\hat{\beta}_{(X,y)}$ that are piece-wise linear functions of $\lambda$ in total $(\lambda_{\max} - \lambda_{\min})/\epsilon$ [26]. Moreover, it is also guaranteed that the error between $\beta_{(X,y)}^{(\epsilon)}$ and $\hat{\beta}_{(X,y)}$ is uniformly upper bounded for all $\lambda \in [\lambda_{\min}, \lambda_{\max}]$.

---

**Algorithm 2** Approximate incremental quadratic algorithm for RLR with $\ell_2$ penalty, [2]

---

Set $\beta_0^{(\epsilon)} = \hat{\beta}_{(X,y)}(\lambda_{\min}), t = 0$.
**while** $\lambda_t < \lambda_{\max}$ **do**
    $\lambda_{t+1} = \lambda_t + \epsilon$
    $\beta^{(\epsilon)}(\lambda) = \beta_t^{(\epsilon)} - \left[\nabla^2 l\left(\beta_t^{(\epsilon)}, (X,y)\right) + 2\lambda_{t+1}I\right]^{-1} \cdot \left[\nabla l\left(\beta_t^{(\epsilon)}, (X,y)\right) + 2\lambda_{t+1}\beta_t^{(\epsilon)}\right]$
    $t = t + 1$

---

**Theorem 4.1** (Theorem 1, [2]). *Given a problem instance $P$, for small enough $\epsilon$, there is a uniform bound $O(\epsilon^2)$ on the error $\|\hat{\beta}_{(X,y)}(\lambda) - \beta_{(X,y)}^{(\epsilon)}(\lambda)\|_2$ for any $\lambda \in [\lambda_{\min}, \lambda_{\max}]$.*

Denote $h_{\text{RLR}}^{(\epsilon)}(\lambda, P) = l_{\text{RLR}}(\beta^\epsilon(\lambda), (X_{\text{val}}, y_{\text{val}}))$ the approximation function of the validation loss $h_{\text{RLR}}(\lambda, P)$. Using Theorem 4.1 and note that the loss $f(z) = \log(1 + e^{-z})$ is 1-Lipschitz, we can show that the difference between $h_{\text{RLR}}^{(\epsilon)}(\lambda, P)$ and $h_{\text{RLR}}(\lambda, P)$ is uniformly upper-bounded.

**Lemma 4.2.** *The approximation error of the validation loss function is uniformly upper-bounded $|h_{RLR}^{(\epsilon)}(\lambda, P) - h_{RLR}(\lambda, P)| = O(\epsilon^2)$, for all $\lambda \in [\lambda_{\min}, \lambda_{\max}]$.*

We now present one of our main results, which is the pseudo-dimension bound of the approximate validation loss function class $\mathcal{H}_{\text{RLR}}^{(\epsilon)}$.

**Theorem 4.3.** *Consider the RLR under $\ell_1$ (or $\ell_2$) constraint with parameter $\lambda \in [\lambda_{\min}, \lambda_{\max}]$ that take a problem instance $P$ drawn from an unknown problem distribution $\mathcal{D}$ over $\Pi_{m,p}$. Under Assumptions 1 and 2, $\mathcal{H}_{RLR}$ is bounded by some constant $H$ (Lemma C.2). Suppose that we use Algorithm 1 (or Algorithm 2) to approximate the solution $\hat{\beta}_{(X,y)}(\lambda)$ by $\beta_{(X,y)}^{(\epsilon)}(\lambda)$ with a uniform error $O(\epsilon^2)$ for any $\lambda \in [\lambda_{\min}, \lambda_{\max}]$, where $\epsilon$ is the approximation step-size. Consider the approximation validation loss function class $\mathcal{H}_{RLR}^{(\epsilon)} = \{h_{RLR}^{(\epsilon)}(\lambda, \cdot) : \Pi_{m,p} \to \mathbb{R}_{\geq 0} \mid \lambda \in [\lambda_{\min}, \lambda_{\max}]\}$, where*

$$h_{RLR}^{(\epsilon)}(\lambda, P) = \frac{1}{m'} \sum_{i=1}^{m'} \log(1 + \exp(-y_i x_i^\top \beta_{(X,y)}^{(\epsilon)}(\lambda)))$$

*is the approximate validation loss. Then we have $\text{Pdim}(\mathcal{H}_{RLR}^{(\epsilon)}) = O(m^2 + \log(1/\epsilon))$. Further, we assume that $\epsilon = O(\sqrt{H})$ where $H$ is the upperbound of $\mathcal{H}_{RLR}$ under Assumptions 1 and 2. Given any set $\mathcal{S}$ of $T$ problem instances drawn from a problem distribution $\mathcal{D}$ over $\Pi_{m,p}$, the empirical Rademacher complexity $\hat{\mathscr{R}}(\mathcal{H}_{RLR}^{(\epsilon)}, \mathcal{S}) = O(H\sqrt{(m^2 + \log(1/\epsilon))/T})$.*

The key observation here is that the approximation solution $\hat{\beta}_{(X,y)}^{(\epsilon)}$ is piece-wise linear over $(\lambda_{\max} - \lambda_{\min})/\epsilon$ pieces, leading to the fact that the approximate validation loss function $h_{\text{RLR}}^{(\epsilon)}(\lambda, \cdot)$ is a "special function" (Pfaffian function [43]) in each piece, which is a combination of exponentiation of linear functions of $\lambda$. The detailed proof of Theorem 4.3 can be found on the Appendix D.3.

## 4.2 Learning guarantees for Regularized Logistic Regression hyperparameter tuning

Our goal now is to use the upper bound for empirical Rademacher complexity of the validation loss function class $\mathcal{H}_{\text{RLR}}$. We use techniques for approximate data-driven algorithm design due to [29], combining the uniform error upper bound between validation loss function $h_{\text{RLR}}(\lambda, P)$ and its approximation $h_{\text{RLR}}^{(\epsilon)}(\lambda, P)$ (Lemma 4.2) and empirical Rademacher complexity of approximation validation loss function class $\mathcal{H}_{\text{RLR}}^{(\epsilon)}$ (Theorem 4.3), to obtain a bound on the empirical Rademacher complexity of $\mathcal{H}_{\text{RLR}}$. This allows us to give a learning guarantee for the regularization parameters $\lambda$, which is formalized by the following theorem.

**Theorem 4.4.** *Consider the RLR under $\ell_1$ (or $\ell_2$) constraint. Under Assumptions 1, 2, $\mathcal{H}_{RLR}$ is bounded by some constant $H$ (Lemma C.2). Consider the class function $\mathcal{H}_{RLR} = \{h_{RLR}(\lambda, \cdot) : \Pi_{m,p} \to \mathbb{R}_{\geq 0} \mid \lambda \in [\lambda_{\min}, \lambda_{\max}]\}$ where $h_{RLR}(\lambda, P)$ is the validation loss corresponding to problem*

*instance $P$ and the $\ell_1$ ($\ell_2$) parameter $\lambda$. Given any set $\mathcal{S}$ of $T$ problem instances drawn from a problem distribution $\mathcal{D}$ over $\Pi_{m,p}$, for any $h_{RLR}(\lambda, \cdot) \in \mathcal{H}_{RLR}$, w.p. $1 - \delta$ for any $\delta \in (0, 1)$, we have*

$$\left| \frac{1}{T} \sum_{i=1}^{T} h_{RLR}(\lambda, P^{(i)}) - \mathbb{E}_{P \sim \mathcal{D}}[h_{RLR}(\lambda, P)] \right| \leq O\left( H\sqrt{\frac{m^2 + \log(1/\epsilon)}{T}} + \epsilon^2 + \sqrt{\frac{1}{T} \log \frac{1}{\delta}} \right),$$

*for some sufficiently small $\epsilon$.*

The proof detail of Theorem 4.4 is included in the Appendix D.3. The above generalization guarantee gives a bound on the average error on RLR validation loss over the problem distribution, for the parameter $\lambda$ learned from $T$ problem instances. In commonly used approaches, the validation set size is small or a constant, and our result can be interpreted as the upper bound on the generalization error in terms of the number of problem instances $T$ and the step length $\epsilon$. We only consider RLR under $\ell_1$ (or $\ell_2$) constraints, which are commonly studied in the literature, our analysis could be easily extended to RLR under $\ell_q$ constraint for any $q \geq 1$.

### 4.3 An extension to 0-1 loss

Since logistic regression is often used for binary classification tasks, it is interesting to consider the 0-1 loss as the validation loss function. It has been shown that $\mathbb{1}\{z \leq 0\} \leq 4\log(1 + e^{-z})$ for any $z$ [44]. This inequality, combined with Theorem 4.4, directly provides a learning guarantee for the 0-1 validation loss function.

**Theorem 4.5.** *Let $\tau > 2\epsilon^2$ and $\delta \in (0, 1)$, where $\epsilon$ is the approximation step-size. Then for any $n \geq s(\tau/2, \delta) = \Omega\left( \frac{H^2(m^2 + \log \frac{1}{\epsilon}) + \log \frac{1}{\delta}}{(\tau/2 - \epsilon^2)^2} \right)$, if we have $n$ problem instances $\{P^{(i)}, \ldots, P^{(n)}\}$ drawn i.i.d. from some problem distribution $\mathcal{D}$ over $\Pi_{m,p}$ to learn the regularization parameter $\lambda^{ERM}$ for RLR via ERM, then*

$$\mathbb{E}_{P \sim \mathcal{D}}(h_{RLR}^{0\text{-}1}(\lambda^{ERM}, P))) \leq 4 \min_{\lambda \in [\lambda_{\min}, \lambda_{\max}]} \mathbb{E}_{P \sim \mathcal{D}}(h_{RLR}(\lambda, P)) + 4\tau.$$

The detailed proof of Theorem 4.5 can be found on Appendix D.4. It is worth noting that we are providing learning guarantee for 0-1 validation loss function class $\mathcal{H}_{RLR}^{0\text{-}1}$ indirectly via the validation loss function class $\mathcal{H}_{RLR}$ with cross-entropy objective function, which is arguably not optimal. The question of how to provide a true PAC-learnable guarantee for $\mathcal{H}_{RLR}^{0\text{-}1}$ remains an interesting challenge.

## 5 Conclusion and future work

In this work, we present novel learning guarantees for tuning regularization parameters for both the Elastic Net and Regularized Logistic Regression models, across problem instances. For the Elastic Net, we propose fine-grained structural results that pertain to the tuning of regularization parameters. We use them to give an improved upper bound on the pseudo-dimension of the relevant validation loss function class of and we prove that our new bound is tight.

For the problem of tuning regularization parameters in regularized logistic regression, we propose an alternative approach that involves analyzing the approximation of the original validation loss function class. This approximation, characterized by a piece-wise linear representation, provides a useful analytical tool in the absence of an exact dependence of the logistic loss on the regularization parameters. Additionally, we employ an upper bound on the approximation error between the original and approximated functions, to obtain a learning guarantee for the original validation loss function class. Remarkably, our proposed approach is not restricted solely to regularized logistic regression but can be extended to a wide range of other problems, demonstrating its generality and applicability.

It is worth noting that this work only focuses on the sample complexity aspect of the hyperparameter tuning in the Elastic Net and Regularized Logistic Regression. The question of computational complexity in this setting is an interesting future direction. Other interesting questions include designing hyperparameter tuning techniques for this setting that are robust to adversarial attacks, and hyperparameter tuning for Regularized Logistic Regression with both $\ell_1$ and $\ell_2$ constraints.

## 6   Acknowledgement

We thank Yingyu Liang for useful discussions during early stages of this work and Mikhail Khodak for helpful feedback. This work was supported in part by NSF grants CCF-1910321 and SES-1919453, the Defense Advanced Research Projects Agency under cooperative agreement HR00112020003, and a Simons Investigator Award.

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
