# A    Classical Generalization Bounds

In this section, we will provide basic terminologies from classical learning theory which will be useful in our analysis.

## A.1    Pseudo-dimension

The pseudo-dimension is frequently used to analyze the learning theoretic complexity of a real-valued function class. The formal definition is stated here for convenience.

**Definition 4** (Shattering and Pseudo-dimension, [33]). *Let $\mathcal{F}$ be a set of functions mapping from $\mathcal{X}$ to $\mathbb{R}$, and suppose that $S = \{x_1, \ldots, x_m\} \subseteq \mathcal{X}$. Then $S$ is pseudo-shattered by $\mathcal{F}$ if there are real numbers $r_1, \ldots, r_m$ such that for each $b \in \{0, 1\}^m$ there is a function $f_b$ in $\mathcal{F}$ with $\mathrm{sign}(f_b(x_i) - r_i) = b_i$ for $i \in [m]$. We say that $r = (r_1, \ldots, r_m)$ witnesses the shattering. We say that $\mathcal{F}$ has pseudo-dimension $d$ if $d$ is the maximum cardinality of a subset $S$ of $\mathcal{X}$ that is pseudo-shattered by $\mathcal{F}$, denoted $\mathrm{Pdim}(\mathcal{F}) = d$. If no such maximum exists, we say that $\mathcal{F}$ has infinite pseudo-dimension.*

The following lemma is particularly useful when we analyze the pseudo-dimension of a function class that is a composition of a monotonic function and another simpler function class. The result is useful in our analysis of regularized logistic regression (Section D).

**Lemma A.1** ([45]). *Suppose $\mathcal{F}$ is a class of real-valued functions and $\sigma : \mathbb{R} \to \mathbb{R}$ is a non-decreasing function. Let $\sigma(\mathcal{F})$ denote the class $\{\sigma \circ f : f \in \mathcal{F}\}$. Then $\mathrm{Pdim}(\sigma(\mathcal{F})) \leq \mathrm{Pdim}(\mathcal{F})$. The equality holds if $\sigma$ is a continuous and strictly increasing function.*

*On other hand, if $\sigma$ is a non-increasing function then $\mathrm{Pdim}(\sigma(\mathcal{F})) \geq \mathrm{Pdim}(\mathcal{F})$. The equality holds if $\sigma$ is a continuous and strictly decreasing function.*

## A.2    (Empirical) Rademacher Complexity

Another tool for analyzing the complexity of a real-valued function is the empirical Rademacher Complexity, which will be defined below.

**Definition 5** (Empirical Rademacher Complexity, [9]). *Let $\mathcal{F}$ be a set of functions mapping from $\mathcal{X}$ to $\mathbb{R}$, and let $S = \{x_1, \ldots, x_T\} \subseteq \mathcal{X}$ be a set of $T$ samples from $\mathcal{X}$. The empirical Rademacher Complexity of $\mathcal{F}$ with respect to $S$ is defined as*

$$\hat{\mathscr{R}}(\mathcal{F}, S) = \mathbb{E}_\sigma \left[ \sup_{f \in \mathcal{F}} \sum_{i=1}^{T} \sigma_i f(x_i) \right],$$

*where $\sigma_i$ is Rademacher random variable for $i \in [T]$.*

## A.3    Uniform Convergence

The following classical result establishes a connection between the uniform convergence and the pseudo-dimension of real-valued function classes.

**Theorem A.2** ([33]). *Suppose $\mathcal{F}$ is a class of real-valued functions with range in $[0, H]$ and finite $\mathrm{Pdim}(\mathcal{F})$. Then for any $\epsilon > 0$ and $\delta \in (0, 1)$, for any distribution $\mathcal{D}$ and for any set $S$ of $m = O\left( \frac{H^2}{\epsilon^2} (\mathrm{Pdim}(\mathcal{F}) + \log \frac{1}{\delta}) \right)$ samples drawn from $\mathcal{D}$, w.p. at least $1 - \delta$, we have*

$$|L_S^m(f) - L_\mathcal{D}(f)| < \epsilon, \quad \text{for all } f \in \mathcal{F}.$$

# B    Lemmas, Proof Details for Section 3

In this section, we provide the details for results discussed in Section 3.

## B.1    Upper bound

We will state results from prior research which are useful in establishing our pseudo-dimension upper bound, followed by full proof details.

### B.1.1 Basic Structural Results About Elastic Net

We first present basic structural results about the Elastic Net. The result allows us to rewrite any Elastic Net problem into an equivalent LASSO problem.

**Lemma B.1** (LASSO reduction of the Elastic Net, [1]). *Given $(X, y) \in \mathbb{R}^{m \times p} \times \mathbb{R}^m$ and $(\lambda_1, \lambda_2) \in (0, \infty) \times [0, \infty)$, define the ElasticNet problem*

$$\min_{\beta \in \mathbb{R}^p} \|y - \beta X\|_2^2 + \lambda_1 \|\beta\|_1 + \lambda_2 \|\beta\|_2^2.$$

*Define the new dataset $(X', y')$*

$$X'_{(n+p) \times p} = (1 + \lambda_2)^{-1/2} \begin{pmatrix} X \\ \sqrt{\lambda_2} I_p \end{pmatrix}, \quad y'_{n+p} = \begin{pmatrix} y \\ 0 \end{pmatrix}.$$

*Let $\gamma = \frac{\lambda_1}{\sqrt{1+\lambda_2}}$. Then the original ElasticNet problem can be written as*

$$\min_{\beta' \in \mathbb{R}^p} \|y' - X'\beta'\|_2^2 + \gamma \|\beta'\|_1.$$

*Let $\hat{\beta} = \operatorname{argmin}_{\beta \in \mathbb{R}^p} \|y - \beta X\|_2^2 + \lambda_1 \|\beta\|_1 + \lambda_2 \|\beta\|_2^2$ and $\hat{\beta}' = \operatorname{argmin}_{\beta' \in \mathbb{R}^p} \|y' - X'\beta'\|_2^2 + \gamma \|\beta'\|_1$, then*

$$\hat{\beta} = \frac{1}{\sqrt{1 + \lambda_2}} \hat{\beta}'.$$

The following result (Lemma B.2) characterizes the solutions of the LASSO problem. To state it, we will need a couple definitions. The *general position* is a standard mild assumption on the design matrix $X$.

**Definition 6** (General position, [27]). *A matrix $X \in \mathbb{R}^{m \times p}$ is said to have its columns in the general position if the affine span of any $k \le m$ points $(\sigma_i x_{j_i})_{i \in [k], j_{i_i} = J \subseteq [p]}$ for arbitrary signs $\sigma_{[k]} \in \{-1, 1\}^k$ and subset $J$ of the columns of size $k$, does not contain any element of $\{x_i | i \notin J\}$.*

Equicorrelation set (sometimes called active set) is the set of covariates with maximum absolute value of correlation for the LASSO fit corresponding to a given value of $\lambda_1$.

**Definition 7** (Equicorrelation sets, [27]). *Let $\hat{\beta} \in \operatorname{argmin}_{\beta \in \mathbb{R}^p} \frac{1}{2} \|y - X\beta\|_2^2 + \lambda_1 \|\beta\|_1$. The equicorrelation set corresponding to $\hat{\beta}$, $\mathcal{E} = \left\{ j \in [p] \mid \left| \boldsymbol{x}_j^\top \left( y - X\hat{\beta} \right) \right| = \lambda_1 \right\}$ is simply the set of covariates with maximum absolute correlation. We also define the equicorrelation sign vector for $\hat{\beta}$ as $s = sign(X_{\mathcal{E}}^\top (y - X\hat{\beta})) \in \{\pm 1\}^{|\mathcal{E}|}$.*

We are now ready to state the unique closed-form solution of the LASSO under general position assumption, in terms of equicorrelation sets.

**Lemma B.2** (Closed-form solution of the LASSO, [27]). *If the columns of $X$ are in general position, then for any $y$ and $\lambda_1 > 0$, the LASSO solution is unique and is given by*

$$\hat{\beta}_{\mathcal{E}} = (X_{\mathcal{E}}^\top X_{\mathcal{E}})^{-1}(X_{\mathcal{E}}^\top y - \lambda_1 s), \quad \hat{\beta}_{[p] \setminus \mathcal{E}} = 0$$

*where $\mathcal{E}$ and $s$ are the equicorrelation set and equicorrelation sign vector corresponding to $\hat{\beta}$.*

Therefore, the solution of Elastic Net can be written as below.

**Lemma B.3** (Closed-form solution of the ElasticNet, [1]). *Let $X$ be the matrix with columns in the general position, and $\lambda = (\lambda_1, \lambda_2) \in \mathbb{R}_{>0}^2$. Then the ElasticNet solution $\hat{\beta}(\lambda) \in \operatorname{argmin}_{\beta \in \mathbb{R}^p} \|y - X\beta\|_2^2 + \lambda_1 \|\beta\|_1 + \lambda_2 \|\beta\|_2$ is unique for any dataset $(X, y)$ and satisfies*

$$(\hat{\beta}(\lambda))_{\mathcal{E}} = \left( X_{\mathcal{E}}^\top X_{\mathcal{E}} + \lambda_2 I_{|\mathcal{E}|} \right)^{-1} X_{\mathcal{E}}^\top y - \lambda_1 \left( X_{\mathcal{E}}^\top X_{\mathcal{E}} + \lambda_2 I_{|\mathcal{E}|} \right)^{-1} s, \quad (\hat{\beta}(\lambda))_{[p] \setminus \mathcal{E}} = 0$$

*for some $\mathcal{E} \in [p]$ and $s \in \{-1, 1\}^p$.*

The following result describes the relation between the solution of Elastic Net and the coefficient parameters $\lambda$. The proof of the result can be easily derived based on simple algebra.

**Lemma B.4** ([14]). *Let $A$ be an $r \times s$ matrix. Consider $B(\lambda) = (A^\top A + \lambda I_s)^{-1}$.*

1. *Each entry of $B(\lambda)$ is a rational polynomial $P_{ij}(\lambda)/Q(\lambda)$ for $i, j \in [s]$ with each $P_{ij}$ of degree at most $s - 1$, and $Q$ of degree $s$.*

2. *Further, for $i = j$, $P_{ij}$ has degree $s - 1$ and leading coefficient 1, and for $i \neq j$, $P_{ij}$ has degree at most $s - 2$. Also, $Q(\lambda)$ has leading coefficient 1.*

### B.1.2 Proofs of Main Theorems

We now give a detailed proof for the main structural results (Theorem 3.2). We start with a useful definition.

**Definition 8** (Semi-algebraic sets, Algebraic curves). *A semi-algebraic sets of $\mathbb{R}^n$ is a finite union of sets of the form $\{x \in \mathbb{R}^n \mid p_i(x) \geq 0, \text{ for } i \in [m]\}$, where $p_i$ are polynomials. An algebraic curve is the zero set of a polynomial in two dimensions.*

We will now restate and prove Theorem 3.2.

**Theorem 3.2 (restated).** *Let $\mathcal{H}_{EN} = \{h_{EN}(\lambda, \cdot) : \Pi_{m,p} \to \mathbb{R}_{\geq 0} \mid \lambda \in \mathbb{R}^2_{>0}\}$ the class of Elastic Net validation loss function class. Consider the dual class $\mathcal{H}^*_{EN} = \{h^*_P : \mathcal{H}_{EN} \to \mathbb{R}_{\geq 0} \mid P \in \Pi_{m,p}\}$, where $h^*_P(h_{EN}(\lambda, \cdot)) = h_{EN}(\lambda, P)$. Then $\mathcal{H}^*_{EN}$ is $(\mathcal{F}, 3^p, \mathcal{G}, p3^p)$-piecewise decomposable, where the piece function class $\mathcal{F} = \{f_q : \mathcal{H}_{EN} \to \mathbb{R}\}$ consists at most $3^p$ rational function $f_{q_1, q_2} : h_{EN}(\lambda, \cdot) \mapsto \frac{q_1(\lambda_1, \lambda_2)}{q_2(\lambda_1, \lambda_2)}$ of degree at most $2p$, and the boundary function class $\mathcal{G} = \{g_r : \mathcal{H}_{EN} \to \{0, 1\}\}$ consists of semi-algebraic sets bounded by at most $p3^p$ algebraic curves $g_r : h_{EN}(\lambda, \cdot) \mapsto \mathbb{1}\{r(\lambda_1, \lambda_2) < 0\}$, where $r$ is a polynomial of degree at most $p$.*

*Proof.* Given a problem instance $P = (X, y, X_{\text{val}}, y_{\text{val}}) \in \Pi_{m,p}$, from Lemma B.3, for each $\lambda$, the solution $\hat{\beta}(\lambda)$ of the Elastic Net can be characterized as follow

$$\hat{\beta}(\lambda) = (X_{\mathcal{E}}^\top X_{\mathcal{E}} + \lambda_2 I_{|\mathcal{E}|})^{-1} X_{\mathcal{E}}^\top y - \lambda_1 (X_{\mathcal{E}}^\top X_{\mathcal{E}} + \lambda_2 I_{|\mathcal{E}|})^{-1} s,$$

for some $\mathcal{E} \in [p]$ and $s \in \{\pm 1\}^p$. Therefore, the prediction $\hat{y}$ on any validation example with features $x \in \mathbb{R}^p$ is

$$\hat{y} = x\hat{\beta}(\lambda) = x[(X_{\mathcal{E}}^\top X_{\mathcal{E}} + \lambda_2 I_{|\mathcal{E}|})^{-1} X_{\mathcal{E}}^\top y - \lambda_1 (X_{\mathcal{E}}^\top X_{\mathcal{E}} + \lambda_2 I_{|\mathcal{E}|})^{-1} s].$$

This implies that: for any region $R \subset \mathbb{R}^2_{>0}$, if the equicorrelation set and sign vector $(\mathcal{E}, s)$ is fixed over $R$, then the solution $\hat{\beta}(\lambda)$ and the prediction $y$ corresponding to $x$ is also fixed. Consequently, within any region $R$ where $(\mathcal{E}, s)$ remains unchanged, Lemma B.4 establishes that the validation loss function $h_{EN}(\lambda, P)$ (associated with a given problem instance $P$) is a constant rational function of the form $\frac{q_1(\lambda_1, \lambda_2)}{q_2(\lambda_1, \lambda_2)}$, where $q_1$ and $q_2$ are polynomials of degree at most $2p$ (since $2|\mathcal{E}| \leq 2p$ by definition). Notably, there are at most $3^p$ distinct values of $(\mathcal{E}, s)$, which implies that $h_{EN}(\lambda, P)$ can take on at most $3^p$ different polynomial forms.

The only remaining task is to examine the semi-algebraic sets and algebraic curves that separates region $R$. Consider such region $R$, in which the equicorrelation set and sign $(\mathcal{E}, s)$ is fixed.

- *Condition for a feature enters $\mathcal{E}$*: consider a feature $j \notin \mathcal{E}$, the condition for $j$ to enters $\mathcal{E}$ is
$$(x_j^*)^\top (y^* - X_{\mathcal{E}}^*(c_1 - c_2 \lambda_1^*)) = \pm \lambda_1^*$$

where $c_1 = (X_{\mathcal{E}}^{*\top} X_{\mathcal{E}}^*)^{-1}$, $c_2 = (X_{\mathcal{E}}^{*\top} X_{\mathcal{E}}^*)^{-1} s$, $X^* = \frac{1}{\sqrt{1+\lambda_2}} \begin{bmatrix} X \\ \sqrt{\lambda_2} I_p \end{bmatrix}$, $y^* = \begin{bmatrix} y \\ 0 \\ \vdots \\ 0 \end{bmatrix}$. Simplifying

the equation above, we have
$$\lambda_1^* - \frac{(x_j^*)^\top X_{\mathcal{E}}^*(X_{\mathcal{E}}^* X_{\mathcal{E}}^*)^{-1}(X_{\mathcal{E}}^*)^\top y^* - (x_j^*)^\top y^*}{(x_j^*)^\top X_{\mathcal{E}}^*(X_{\mathcal{E}}^{*\top} X_{\mathcal{E}}^*)^{-1} s \pm 1} = 0, \text{ or}$$
$$\lambda_1 (x_j^\top (X_{\mathcal{E}} X_{\mathcal{E}}^{*\top})^{-1} X_{\mathcal{E}} s \pm 1) - x_j^\top X_{\mathcal{E}} (X_{\mathcal{E}}^\top X_{\mathcal{E}} + \lambda_2 I_{|\mathcal{E}|})^{-1} X_{\mathcal{E}}^\top y - x_j^\top y = 0,$$

which is an algebraic curve with the RHS is a polynomial of degree at most $p$.

- *Condition for a feature leaves $\mathcal{E}$*: consider a feature $j' \in \mathcal{E}$. Similar to the previous case, the condition for $j'$ to leave $\mathcal{E}$ can be described by an algebraic curve with the RHS as a polynomial of degree at most $p$.

Finally, notice that there are at most $\sum_{i=0}^{p} \binom{p}{i}((p-i)+i) = p3^p$ curves, across which the equicorrelation set and sign $(\mathcal{E}, s)$ might change, which concludes the proof. $\qquad\square$

Using the GJ framework (Theorem 3.1), one can show that if a function class $\mathcal{H}$ has its dual-class $\mathcal{H}^*$ is piece-wise decomposable (in the sense of Definition 3), and all the piece and boundary functions are rational functions with upper bounded degree, then $\mathrm{Pdim}(\mathcal{H})$ is upper bounded.

**Lemma B.5.** *Consider the function class $\mathcal{H} = \{h(a, \cdot) : \mathcal{X} \to \mathbb{R} \mid a \in \mathbb{R}^n\}$ be a function class parameterized by $a \in \mathbb{R}^W$. Consider the dual class $\mathcal{H}^* = \{h_x(\cdot) : \mathbb{R}^n \to \mathbb{R} \mid x \in \mathcal{X}\}$, where $h_x(a) = h(a, x)$. Assume that $\mathcal{H}^*$ is $(\mathcal{F}, k_{\mathcal{F}}, \mathcal{G}, k_{\mathcal{G}})$ piece-wise decomposable, and $\mathcal{F}, \mathcal{G}$ contains only rational functions in $a$ of degree at most $\Delta$. Then $\mathrm{Pdim}(\mathcal{F}) = O(n \log(\Delta(k_{\mathcal{F}} + k_{\mathcal{G}})))$.*

*Proof.* Given an input $x \in \mathcal{X}$ and a threshold $t \in \mathbb{R}$, for any function $h(a, \cdot) \in \mathcal{H}$ corresponding to parameter $a$, consider the computation $\Gamma_{x,t} : \mathcal{H} \to \{0, 1\}$, where

$$\Gamma_{x,t}(h(a, \cdot)) = \mathbb{1}\{h(a, x) - t \geq 0\}, \text{ for any } h(a, \cdot) \in \mathcal{H}.$$

Our goal now is to show that $\Gamma_{x,t}$ is a GJ algorithm in the sense of Definition 1.

From assumptions, we know that the dual class $\mathcal{H}^*$ is $(\mathcal{F}, k_{\mathcal{F}}, \mathcal{G}, k_{\mathcal{G}})$ piece-wise decomposable, where $\mathcal{F}, \mathcal{G}$ consists of rational function in $a$ of degree at most $\Delta$. This implies that for any $h(a, \cdot) \in \mathcal{H}$, the function $h_x(a) = h(a, x)$ is a rational function of $a$, of which the form is one of $k_{\mathcal{F}}$ rational functions in $\mathcal{F}$. Hence, to compute $\Gamma_{x,t}(h(a, \cdot))$, one needs to specify the closed-form of $h(a, \cdot)$, which is determined by binary-valued vector $b_a = \{g^{(1)}(a), \ldots, g^{(k_{\mathcal{G}})}(a)\}$, and can be calculated as conditional statements in the form $\mathbb{1}\{g^{(i)}(a) \geq 0\}$ for $i \in k_{\mathcal{G}}$. Therefore, we conclude that the computation of $\Gamma_{x,t}$ can be described by a GJ algorithm.

The predicate complexity of $\Gamma_{x,t}$ is the total number of functions in $\mathcal{F}$ and $\mathcal{G}$, which is equal to $k_{\mathcal{F}} + k_{\mathcal{G}}$. The degree of $\Gamma_{x,t}$ is the maximum degree of rational functions in $\mathcal{F}$ and $\mathcal{G}$, which is $\Delta$ from assumptions. From Theorem 3.1, we conclude that $\mathrm{Pdim}(\mathcal{F}) = O(n \log(\Delta(k_{\mathcal{F}} + k_{\mathcal{G}}))$. $\qquad\square$

**Theorem 3.3.** *Let $\mathcal{H}_{EN} = \{h_{EN}(\lambda, \cdot) : \Pi \to \mathbb{R}_{\geq 0} \mid \lambda \in \mathbb{R}_{>0}^2\}$ be the Elastic Net validation loss function class that maps problem instance $P$ to validation loss $\ell_{val}(\lambda, P)$. Then $\mathrm{Pdim}(\mathcal{H}_{EN})$ is $O(p)$.*

*Proof.* Given a problem instance $P \in \Pi_{m,p}$ and a threshold $t \in \mathbb{R}$, for any validation loss function $h_{\mathrm{EN}}(\lambda, \cdot) \in \mathcal{H}_{\mathrm{EN}}$, consider the computation $\Gamma_{P,t} : \mathcal{H}_{\mathrm{EN}} \to \{0, 1\}$, where

$$\Gamma_{P,t}(h(\lambda, \cdot)) = \mathbb{1}\{h(\lambda, P) - t \geq 0\}, \text{ for any } h(\lambda, \cdot) \in \mathcal{H}_{\mathrm{EN}}.$$

From Theorem 3.2, for a given problem instance $P$, we know that the dual-class $\mathcal{H}_{\mathrm{EN}}^*$ is $(\mathcal{F}, 3^p, \mathcal{G}, p3^p)-$piecewise decomposable, where $\mathcal{F}$ consists at most $3^p$ rational function of degree at most $2p$, and $\mathcal{G}$ consists of at most $p3^p$ algebraic curves of degree at most $p$. From Lemma B.5, $\mathrm{Pdim}(\mathcal{H}_{\mathrm{EN}}) = O(2 \log(2p(p+1)3^p) = O(p)$.

$\qquad\square$

## B.2 Lower bound

We now instantiate a formal proof for Theorem 3.5.

**Theorem 3.5 (restated).** *Let $\mathcal{H}_{LASSO}$ be a set of functions $\{h_{LASSO}(\lambda, \cdot) : \Pi_{m,p} \to \mathbb{R}_{\geq 0} \mid \lambda \in \mathbb{R}^+\}$ that map a regression problem instance $P \in \Pi_{m,p}$ to the validation loss $h_{LASSO}(\lambda, P)$ of LASSO trained with regularization parameter $\lambda$. Then $\mathrm{Pdim}(\mathcal{H}_{LASSO})$ is $\Omega(p)$.*

*Proof.* Our proof of the lower bound in Theorem 3.5 builds on the "adversarial strategy" due to [35], where a data set $(X, y)$ is constructed with the largest possible number of segments in the LASSO regularization path, for any $p$. Here we will include and discuss the main results from [35] that are useful in understanding our proof.

Our approach is to construct $N = p$ problem instances such that all $2^N$ above-below patterns (w.r.t. witness values) for the validation loss are achieved by choosing appropriate points ($\lambda$ values) on the piecewise linear regularization path of the training instance, by utilizing the property that all unsigned sparsity patterns are achieved by the construction of [35]. In more detail, recall that the *signed* sparsity pattern $\{\eta_1, \ldots, \eta_k\}$ of a piecewise-linear regularization path $P$ for dataset $(X, y)$ is a sequence of vectors in $\{\pm 1, 0\}^p$ corresponding to the signs of the coefficients of the LASSO fit $\hat{\beta}^{(X,y)}(\lambda)$ in consecutive pieces of $P$, i.e. $\eta_j = (\text{sign}(\hat{\beta}_i^{(X,y)}(\lambda_j)))_{i=1}^p$ where $\lambda_j$ corresponds to an interior point of the $j$-th piece of $P$. Let's further denote by $U_P = \{\bar{\eta}_j \mid 1 \le j \le k\}$ where $\bar{\eta}_j = (|\eta_{j1}|, \ldots, |\eta_{jp}|) \in \{0, 1\}^p$ as the *unsigned* sparsity pattern of path $P$.

We use the same training set $(X, y)$ (but different validation sets) across our problem instances, namely the one with $(3^p + 1)/2$ segments constructed by Mairal and Yu (Theorem 1 of [35]). A useful property of this problem instance is that it achieves all the unsigned sparsity patterns, which follows from the following proposition.

**Proposition B.6** ([35]). *Consider $y$ in $\mathbb{R}^n$ and $X$ in $\mathbb{R}^{n \times p}$ such that $X_{\mathcal{E}}$ is full rank for each $\mathcal{E} \subseteq [p]$ and $y$ is in the span of $X$. Denote by $P$ the regularization path of the Lasso problem corresponding to $(X, y)$, and by $k$ the number of linear segments of $P$. Then, there exist $y'$ in $\mathbb{R}^{n+1}$ and $X'$ in $\mathbb{R}^{(n+1) \times (p+1)}$ such that the regularization path $P'$ of the Lasso problem associated to $(X', y')$ has $3k - 1$ linear segments. Moreover, let $\{\eta_1 = 0, \eta_2, \ldots, \eta_k\}$ denote the sequence of sparsity patterns in $\{-1, 0, 1\}^p$ of $P$ (the coordinate-wise signs of the solutions $\hat{\beta}^{(X,y)}(\lambda)$), ordered from large to small values of $\lambda$. The sequence of sparsity patterns in $\{-1, 0, 1\}^{p+1}$ of the new path $P'$ is the following:*

$$\left\{ \begin{bmatrix} \eta_1 \\ 0 \end{bmatrix}, \begin{bmatrix} \eta_2 \\ 0 \end{bmatrix}, \ldots, \begin{bmatrix} \eta_k \\ 0 \end{bmatrix}, \begin{bmatrix} \eta_k \\ 1 \end{bmatrix}, \begin{bmatrix} \eta_{k-1} \\ 1 \end{bmatrix}, \ldots, \begin{bmatrix} \eta_1 = 0 \\ 1 \end{bmatrix}, \begin{bmatrix} -\eta_2 \\ 1 \end{bmatrix}, \ldots, \begin{bmatrix} -\eta_k \\ 1 \end{bmatrix} \right\}.$$

Formally, one could use a simple inductive argument to establish the above claim. In the base case ($p = 1$), $X = y = [1]$ and it is easy to verify that the regularization path $P_1$ consists of two segments with $U_{P_1} = \{0, 1\}$. In the inductive case ($p + 1$ features), consider the first $2k$ sign patterns for the path $P'$ in Proposition B.6. Using the inductive hypothesis, it is readily verified that the number of unsigned sparsity patterns in the regularization path $P'$ is $|U_{P'}| = 2|U_P| = 2^{p+1}$.

In other words, all subsets of the $p$ features appear as "active sets" of coefficients along the regularization path of the training set $(X, y)$. By carefully setting the validation sets across the $p$ problem instances in our proof of Theorem 3.5, we are able to ensure that the validation loss is non-zero exactly in the subset of problems corresponding to the unsigned sparsity patterns of $\hat{\beta}^{(X,y)}(\lambda)$. Thus, the property that all $2^p$ unsigned sparsity patterns are achieved for certain values of $\lambda$ implies that all $2^N$ validation loss patterns are achieved w.r.t. witnesses $0^p$. $\qquad \square$

## C  Boundedness results for validation loss function classes of Elastic Net and Regularized Logistic Regression

In this section, we will give a formal guarantee for the boundedness of the validation loss function class of Elastic Net $\mathcal{H}_{\text{EN}}$ and Regularized Logistic Regression $\mathcal{H}_{\text{RLR}}$, which is essential for establishing learning guarantees for both function classes.

### C.1  Boundedness of the validation loss function class of Elastic Net

The following lemma essentially shows that under mild assumptions on the value of data and the search space of hyperparameters, the validation loss function class $\mathcal{H}_{\text{EN}}$ is uniformly bounded by some constant $H > 0$.

**Lemma C.1.** *Under Assumptions 1 and 2, there exists a uniform constant $H > 0$ so that for all $h_{EN}(\lambda, \cdot) \in \mathcal{H}_{EN} = \{h_{EN}(\lambda, \cdot) : \Pi_{m,p} \to \mathbb{R}_{\ge 0} \mid \lambda \in [\lambda_{\min}, \lambda_{\max}]\}$, we have $\|h_{EN}(\lambda, \cdot)\|_{\infty} = \sup_{P \in \Pi_{m,p}} |h_{EN}(\lambda, P)| \le H$.*

*Proof.* For any problem instance $P = (X, y, X_{\text{val}}, y_{\text{val}}) \in \Pi_{m,p}$, and for any $\lambda = (\lambda_1, \lambda_2) \in [\lambda_{\min}, \lambda_{\max}]^2$, consider the optimization problem for training set

$$\underset{\beta}{\arg\min} \, F(\beta), \qquad (3)$$

where $F(\beta) = \frac{1}{2m} \|y - X\beta\|_2^2 + \lambda_1 \|\beta\|_1 + \lambda_2 \|\beta\|_2^2$. If we set $\beta = \vec{0}$, we have

$$F(\vec{0}) = \frac{1}{2m} \|y\|_2^2 \le C,$$

for some constant $C$ that only depends on $R_2$, due to Assumption 2. Let $\hat{\beta}_{(X,y)}(\lambda)$ be the optimal solution of 3, we have

$$C \ge F(\hat{\beta}_{(X,y)}(\lambda)) \ge \lambda_1 \left\|\hat{\beta}_{(X,y)}(\lambda)\right\|_1 + \lambda_2 \left\|\hat{\beta}_{(X,y)}(\lambda)\right\|_2^2.$$

Therefore, for any problem instance $P$, the solution of the training optimization problem $\hat{\beta}_{(X,y)}(\lambda)$ has bounded norm, i.e. $\left\|\hat{\beta}_{(X,y)}(\lambda)\right\|_1, \left\|\hat{\beta}_{(X,y)}(\lambda)\right\|_2^2 \le \frac{C}{\lambda_{\min}}$, which implies

$$h_{\mathrm{EN}}(\lambda, P) = \frac{1}{2m} \left\|y_{\mathrm{val}} - \hat{\beta}_{(X,y)}(\lambda) X_{\mathrm{val}}\right\|_2^2 \le \frac{1}{2m} \|y_{\mathrm{val}}\|_2^2 + \frac{1}{2m} \left\|\hat{\beta}_{(X,y)}(\lambda) X_{\mathrm{val}}\right\|_2^2 \le H,$$

for some constant $H$ (that only depends on $R_1$, $R_2$ and $\lambda_{\min}$). $\qquad\square$

## C.2 Boundedness of the validation loss function class of Regularized Logistic Regression

Using similar argument, we also have the following claim for the boundedness of validation loss function class of Regularized Logistic Regression.

**Lemma C.2.** *There exists a uniform constant $H > 0$ so that for all $h_{RLR}(\lambda, \cdot) \in \mathcal{H}_{RLR} = \{h_{RLR}(\lambda, \cdot) : \Pi_{m,p} \to \mathbb{R}_{\ge 0} \mid \lambda \in [\lambda_{\min}, \lambda_{\max}]\}$, we have $\|h_{RLR}(\lambda, \cdot)\|_\infty = \sup_{P \in \Pi_{m,p}} |h_{RLR}(\lambda, P)| \le H$.*

# D   Lemmas and Proof Details for Section 4

In this section, we present the detailed proofs of main results in Section 4.

## D.1   Connected Components and Classical Results

We first present some classical results which are useful for analyzing the approximation validation loss function class $\mathcal{H}_{\mathrm{RLR}}^{(\epsilon)}$.

We recall a well-known notion to analyze the pseudo-dimension of a function class, called the *solution set components bound* [33]. The bound on the solution set components essentially refers to the largest number of connected components within the parameter space of a parameterized function class $\mathcal{F}$. This component is generated from the solution set of a system of equations, corresponding to zero sets of functions in $\mathcal{F}$.

**Definition 9** ([33]). *Let $\mathcal{F}$ be a set of real-valued functions defined on $\mathbb{R}^W$. We say that $\mathcal{F}$ has solution set components bound $B$ if for any $1 \le K \le W$ and any $\{f_1, \ldots, f_K\} \subseteq \mathcal{F}$ that has regular zero-set intersections, we have*

$$\max_{K \le W} CC \left( \bigcap_{i=1}^K \{a \in \mathbb{R}^W : f_i(a) = 0\} \right) = B$$

*where $CC(X)$ is the number of connected components of $X$.*

Let us now introduce a definition for the growth function of a binary-valued class function $\mathcal{H}$. This concept essentially quantifies the maximum number of distinct sign patterns $\{h(x_1), \ldots, h(x_m)\}$ that can be observed when we vary the function $h$ across $\mathcal{H}$, considering a set of data points $x_1, \ldots, x_m$.

**Definition 10** (Growth function, [33]). *Given $m$ samples $x_1, \ldots, x_m \in \mathcal{X}$ and let $S = \{x_1, \ldots, x_m\}$. Consider a class function $\mathcal{H}$, of which each $h \in \mathcal{H}$ is a function from $\mathcal{X}$ to $\{-1, 1\}$, and let*

$$\mathcal{H}_S = \{(h(x_1), \ldots, h(x_m)) : h \in \mathcal{H}\}$$

*is the total number of possible ways that $S$ can be classified by $\mathcal{H}$. Then the growth function $G_\mathcal{H}(m)$ is defined as*

$$G_\mathcal{H}(m) = \sup_{x_1, \ldots, x_m} |\mathcal{H}_S|$$

The next classical result establishes a connection between the growth function and the solution set components bound.

**Theorem D.1** (Growth function bound, [33]). *Suppose that $\mathcal{F}$ is a class of real-valued functions defined on $\mathbb{R}^W \times \mathcal{X}$, and that $\mathcal{H}$ is defined as $\{sgn(f) : f \in \mathcal{F}\}$. If $\mathcal{F}$ is closed under addition of constants, has solution set components bound $B$, and functions in $\mathcal{F}$ are $C^W$ in their parameters, then*

$$G_{\mathcal{H}}(m) \le B \left( \frac{em}{W} \right)^W$$

*for $m \ge W$.*

## D.2 The Empirical Rademacher Complexity of Approximate Logistic Validation Loss

We first restate important properties of the approximation solution $\beta^{(\epsilon)}(\lambda)$ accquired using Algorithm 1 (2) in RLR with $\ell_1$ ($\ell_2$) constraint.

**Theorem 4.1 (restated)** ([26]). *Given a problem instance $P = (X, y, X_{val}, y_{val}) \in \Pi_{m,p}$, for small enough $\epsilon$, if we use Algorithm 1 (2) to approximate the solution $\hat{\beta}_{(X,y)}(\lambda)$ of RLR under $\ell_1$ ($\ell_2$) constraint by $\beta^{(\epsilon)}_{(X,y)}(\lambda)$ then there is a uniform bound $O(\epsilon^2)$ on the error $\|\hat{\beta}_{(X,y)}(\lambda) - \beta^{(\epsilon)}_{(X,y)}(\lambda)\|_2$ for any $\lambda \in [\lambda_{\min}, \lambda_{\max}]$.*

*For any $\lambda \in [\lambda_t, \lambda_{t+1}]$, where $\lambda_k = \lambda_{\min} + k\epsilon$, the approximate solution $\beta^{(\epsilon)}(\lambda)$ is calculated by*

$$\beta^{(\epsilon)}_{(X,y)}(\lambda) = \beta^{(\epsilon)}_t - \left[ \nabla^2 l \left( \beta^{(\epsilon)}_t, (X, y) \right)_{\mathcal{A}} \right]^{-1} \cdot \left[ \nabla l \left( \beta^{(\epsilon)}_t, (X, y) \right)_{\mathcal{A}} + \lambda \operatorname{sgn} \left( \beta^{(\epsilon)}_t \right)_{\mathcal{A}} \right] = a_t \lambda + b_t,$$

*if we use Algorithm 1 for RLR under $\ell_1$ constraint, or*

$$\beta^{(\epsilon)}_{(X,y)}(\lambda) = \beta^{(\epsilon)}_t - \left[ \nabla^2 l \left( \beta^{(\epsilon)}_t, (X, y) \right) + 2\lambda_{t+1} I \right]^{-1} \cdot \left[ \nabla l \left( \beta^{(\epsilon)}_t, (X, y) \right) + 2\lambda \beta^{(\epsilon)}_t \right] = a'_t \lambda + b'_t,$$

*if we use Algorithm 2 for RLR under $\ell_2$ constraint.*

The uniform error bound in Theorem 4.1 directly implies the error bound between he validation loss function $h^{(\epsilon)}_{\text{RLR}}(\lambda, P)$ and its approximation $h^{(\epsilon)}_{RLR}(\lambda, P)$. As in prior work [2], we will omit dependence on $\lambda_{\min}, \lambda_{\max}, R$ in our asymptotic upper bounds below.

**Lemma 4.2 (restated).** *The approximation error of the validation loss function is uniformly upper-bounded*

$$|h^{(\epsilon)}_{RLR}(\lambda, P) - h_{RLR}(\lambda, P)| = O(\epsilon^2) \text{, for all } \lambda \in [\lambda_{\min}, \lambda_{\max}].$$

*Proof.* Using triangle inequality and the 1-Lipschitzness of $\mathcal{L}_{log}(z) := \log(1 + e^{-z})$ we have that

$$\left| h^{(\epsilon)}_{\text{RLR}}(\lambda, P) - h_{\text{RLR}}(\lambda, P) \right| = \frac{1}{m'} \left| \sum_{i=1}^{m'} [\mathcal{L}_{log}(y_i x_i^\top \beta^{(\epsilon)}_{(X,y)}(\lambda)) - \mathcal{L}_{log}(y_i x_i^\top \hat{\beta}_{(X,y)}(\lambda))] \right|$$

$$\le \frac{1}{m'} \sum_{i=1}^{m'} \left| \mathcal{L}_{log}(y_i x_i^\top \beta^{(\epsilon)}_{(X,y)}(\lambda)) - \mathcal{L}_{log}(y_i x_i^\top \hat{\beta}_{(X,y)}(\lambda)) \right|$$

$$\le \frac{1}{m'} \sum_{i=1}^{m'} \left| y_i x_i^\top \beta^{(\epsilon)}_{(X,y)}(\lambda) - y_i x_i^\top \hat{\beta}_{(X,y)}(\lambda) \right|.$$

Using Hölder's inequality, Assumption 1, and Theorem 4.1, for any $\lambda \in [\lambda_{\min}, \lambda_{\max}]$, we further have

$$
\begin{aligned}
\left| h_{\mathrm{RLR}}^{(\epsilon)}(\lambda, P) - h_{\mathrm{RLR}}(\lambda, P) \right| &\leq \frac{1}{m'} \sum_{i=1}^{m'} \left| y_i x_i^\top \beta_{(X,y)}^{(\epsilon)}(\lambda) - y_i x_i^\top \hat{\beta}_{(X,y)}(\lambda) \right| \\
&= \frac{1}{m'} \sum_{i=1}^{m'} \left| y_i x_i^\top \left( \beta_{(X,y)}^{(\epsilon)}(\lambda) - \hat{\beta}_{(X,y)}(\lambda) \right) \right| \\
&\leq \frac{1}{m'} \sum_{i=1}^{m'} \|x_i\|_2 \left\| \hat{\beta}_{(X,y)}(\lambda) - \beta_{(X,y)}^{(\epsilon)}(\lambda) \right\|_2 \\
&\leq \left\| \hat{\beta}_{(X,y)}(\lambda) - \beta_{(X,y)}^{(\epsilon)}(\lambda) \right\|_2 \max_i \|x_i\|_2 \\
&= O(\epsilon^2).
\end{aligned}
$$

$\square$

### D.3 Learning the Regularization Hyperparameter in Logistic Regression

In this section, we give a formal proof for Theorem 4.3. We begin by revisiting a fundamental result that proves invaluable when examining function classes that incorporate exponential functions.

**Lemma D.2** ([43]). *Let $Q_i$ ($i \leq m$) be elements of the polynomial ring $\boldsymbol{R}[y_1, \ldots, y_l, e^{\Lambda_1}, \ldots, e^{\Lambda_q}]$, where $\Lambda_i$ are linear function of $y_1, \ldots, y_l$. Suppose that the system*

$$Q_1 = \cdots = Q_m = 0$$

*is regular for $m \leq l$. If $Q_i$ has degree at most $d$ (in $y_1, \ldots, y_l, e^{\Lambda_1}, \ldots, e^{\Lambda_q}$), then the system above has the connected components bound*

$$B_M = 2^{q(q-1)/2} d^l [(l+1)(d+1)]^{l+q}.$$

We now present the formal proof of Theorem 4.3.

**Theorem 4.3 (restated).** *Consider the RLR under $\ell_1$ (or $\ell_2$) constraint with parameter $\lambda \in [\lambda_{\min}, \lambda_{\max}]$ that take a problem instance $P$ drawn from an unknown problem distribution $\mathcal{D}$ over $\Pi_{m,p}$ under Assumptions 1 and 2. Consider the approximation validation loss function class $\mathcal{H}_{RLR}^{(\epsilon)} = \{ h_{RLR}^{(\epsilon)}(\lambda, \cdot) : \Pi_{m,p} \to \mathbb{R}_{\geq 0} \mid \lambda \in [\lambda_{\min}, \lambda_{\max}] \}$, where*

$$h_{RLR}^{(\epsilon)}(\lambda, P) = \frac{1}{m'} \sum_{i=1}^{m'} \log(1 + \exp(-y_i x_i^\top \beta_{(X,y)}^{(\epsilon)}(\lambda)))$$

*is the approximate validation loss. Then we have $\mathrm{Pdim}(\mathcal{H}_{RLR}^{(\epsilon)}) = O(m^2 + \log(1/\epsilon))$. Given any set $\mathcal{S}$ of $T$ problem instances drawn from a problem distribution $\mathcal{D}$ over $\Pi_{m,p}$, the empirical Rademacher complexity $\hat{\mathscr{R}}(\mathcal{H}_{RLR}^{(\epsilon)}, \mathcal{S}) = O(H\sqrt{(m^2 + \log(1/\epsilon))/T})$, where $H$ is the upperbound of original validation loss function class $\mathcal{H}_{RLR}$ (under Assumptions 1 and 2)..*

*Proof.* The proof consists of following steps.

**Step 1:** Simplifying the analysis of $\mathcal{H}_{\mathrm{RLR}}^{(\epsilon)}$ by considering an alternative function class.

Consider any $h(\lambda, \cdot) \in \mathcal{H}_{\mathrm{RLR}}^{(\epsilon)}$, by definition, we have

$$h(\lambda, P) = \frac{1}{m'} \sum_{i=1}^{m'} \log(1 + \exp(-y_i x_i^\top \beta_{(X,y)}^{(\epsilon)}(\lambda)) = \frac{1}{m'} \log \left( \prod_{i=1}^{m'} (1 + \exp(-y_i x_i^\top \beta_{(X,y)}^{(\epsilon)}(\lambda))) \right).$$

From Lemma A.1 and note that $\log(\cdot)$ is a strictly monotonic, continuous function, analyzing the pseudo-dimension of $\mathcal{H}_{\mathrm{RLR}}^{(\epsilon)}$ is equivalent to analyzing the VC-dimension of the class function $\mathcal{G}_{\mathrm{RLR}} = \{ \mathrm{sign}(g_\lambda) : \Pi_{m,p} \times \mathbb{R} \to \{-1, 1\} \mid \lambda \in [\lambda_{\min}, \lambda_{\max}] \}$, where

$$g_\lambda(P, \tau) = \prod_{i=1}^{m'} (1 + \exp(-y_i x_i^\top \beta_{(X,y)}^{(\epsilon)}(\lambda))) - \tau, \tag{4}$$

where $\tau$ is a new variable.

**Step 2:** Using the piece-wise linear property of approximation solution $\beta_{(X,y)}^{(\epsilon)}(\lambda)$, bounding the number of distinct sign patterns of $\{\text{sign}(g_\lambda(P^{(1)}, \tau_1)), \ldots, \text{sign}(g_\lambda(P^{(N)}, \tau_N)\}$, where $(P^{(i)}, \tau_i) \in \Pi_{m,p} \times \mathbb{R}$ for $i \in [N]$, when varying $\lambda \in [\lambda_t, \lambda_{t+1}]$.

Consider $N$ problem instances $\{P^{(1)}, \ldots P^{(N)}\}$ where $P^{(i)} \in \Pi_{m,p}$ for $i \in [N]$, and $N$ corresponding thresholds $\tau_1, \ldots, \tau_N$, to bound the number of distinct sign patterns $\{\text{sign}(g_\lambda(P^{(1)}, \tau_1)), \ldots, \text{sign}(g_\lambda(P^{(N)}, \tau_N))\}$ when varying $\lambda \in [\lambda_t, \lambda_{t+1}]$, we can use Theorem D.1 and bound the solution set components bound $B$, where

$$B = \max_{K \le 1} CC \left( \bigcap_{i=1}^{K} \{\lambda \in [\lambda_t, \lambda_{t+1}] : g_i(\lambda)\} \right),$$

where $g_i(\lambda) = g_\lambda(P^{(i)}, \tau_i)$. From Theorem 4.3 (restated), $\beta_{(X^{(i)}, y^{(i)})}^{(\epsilon)}(\lambda)$ is a linear function of $\lambda$, which implies $g_i(\lambda)$ ($i \in [1]$) is element of polynomials ring $\boldsymbol{R}[\lambda, e^{\Lambda_1}, \ldots, e^{\Lambda_q}]$, where $\Lambda_j$ is linear function of $\lambda$ for $j \in [q]$.

Since $P^{(i)} \in \Pi_{m,p}$, we have $m'_i \le m$ where $m'_i$ is the number of validation sample in validation set of $P^{(i)}$, which implies there is at most $m$ function $\Lambda_j$ (see Eq. 4). Also from Eq. 4, we can see that $g_i(\lambda)$ is a polynomial (in $[\lambda, e^{\Lambda_1}, \ldots, e^{\Lambda_q}]$) of degree at most $m$.

Following Lemma D.2, we conclude that $B \le 2^{m(m-1)/2} m(2(m+1))^{m+1}$. Combining with Theorem D.1, we have the number of distinct sign patterns $\{\text{sign}(g_\lambda(P^{(1)}, \tau_1)), \ldots, \text{sign}(g_\lambda(P^{(N)}, \tau_N))\}$ is upper bounded by $2^{m(m-1)/2} m(2(m+1))^{m+1} \left(\frac{eN}{2}\right)^2$.

**Step 3:** Bounding the pseudo-dimension of $\mathcal{H}_{\text{RLR}}^{(\epsilon)}$.

Note that there are total $(\lambda_{\max} - \lambda_{\min})/\epsilon$ pieces in which $\beta^{(\epsilon)}(\lambda)$ is linear function of $\lambda$. Therefore, using result from **Step 2**, the number of distinct sign patterns $\{\text{sign}(g_\lambda(\tau_1, P^{(1)})), \ldots, \text{sign}(g_\lambda(\tau_N, P^{(N)}))\}$ when varying $\lambda \in [\lambda_{\min}, \lambda_{\max}]$ is upper bounded by $\frac{\lambda_{\max} - \lambda_{\min}}{\epsilon} 2^{m(m-1)/2} m(2(m+1))^{m+1} \left(\frac{eN}{2}\right)^2$. Solving the inequality

$$2^N \le \frac{\lambda_{\max} - \lambda_{\min}}{\epsilon} 2^{m(m-1)/2} m(2(m+1))^{m+1} \left(\frac{eN}{2}\right)^2$$

implies $N = O(m^2 + \log(1/\epsilon))$, which means $\text{Pdim}(\mathcal{H}_{\text{RLR}}) = \text{VCdim}(\mathcal{G}_{\text{RLR}}) = O(m^2 + \log(1/\epsilon))$.

**Step 4:** Bounding the empirical Rademacher complexity of $\mathcal{H}_{\text{RLR}}^{(\epsilon)}$ over a set $\mathcal{S}$ of $T$ problem instances. First, note that under Assumptions 1 and 2, there exists a universal upperbound $H$ for the original validation loss function class $\mathcal{H}_{\text{RLR}}$. Combining with Lemma 4.2, we have the upperbound of $H + O(\epsilon^2)$ for the approximation loss function class $\mathcal{H}_{\text{RLR}}^{(\epsilon)}$.

Using result from **Step 3** and note that $\hat{\mathscr{R}}(\mathcal{H}_{\text{RLR}}^{(\epsilon)}, \mathcal{S}) = O((H + O(\epsilon^2))\sqrt{\text{Pdim}(\mathcal{H}_{\text{RLR}}^{(\epsilon)})/T})$, and $\epsilon = o(\sqrt{H})$, we concludes that $\hat{\mathscr{R}}(\mathcal{H}_{\text{RLR}}^{(\epsilon)}, \mathcal{S}) = O(H\sqrt{(m^2 + \log(1/\epsilon))/T})$ for any set $\mathcal{S}$ of $T$ problem instances drawn from problem distribution $\mathcal{D}$ over $\Pi_{m,p}$. $\qquad\square$

We now give a detailed proof for Theorem 4.4, which establishes the learning guarantee for the validation loss function class $\mathcal{H}_{\text{RLR}}$. We first recall an useful result by Balcan et al. [29], which allows us to upper bound the empirical Rademacher complexity of validation loss function class $\mathcal{H}_{\text{RLR}}$ via that of its approximation $\mathcal{H}_{\text{RLR}}^{(\epsilon)}$.

**Theorem D.3** ([29]). *Let $\mathcal{F} = \{f_r \mid r \in \mathcal{R}\}$ and $\mathcal{G} = \{g_r \mid r \in \mathcal{R}\}$ consist of function mapping $\mathcal{X}$ to $[0, 1]$. For any $\mathcal{S} \subseteq \mathcal{X}$, $\hat{\mathscr{R}}(\mathcal{F}, \mathcal{S}) \le \hat{\mathscr{R}}(\mathcal{G}, \mathcal{S}) + \frac{1}{|\mathcal{S}|} \sum_{x \in \mathcal{S}} \|f_x^* - g_x^*\|$.*

**Theorem 4.4 (restated).** *Consider the class function $\mathcal{H}_{RLR} = \{h_{RLR}(\lambda, \cdot) : \Pi_{m,p} \to \mathbb{R}_{\ge 0} \mid \lambda \in [\lambda_{\min}, \lambda_{\max}]\}$ where $h_{RLR}(\lambda, P)$ is the validation loss corresponding to problem instance $P$ and the $\ell_1$ ($\ell_2$) parameter $\lambda$. Under Assumptions 1 and 2, there exists a universal upperbound for the*

*validation loss function class* $\mathcal{H}_{RLR}$. *Given any set* $\mathcal{S}$ *of* $T$ *problem instances drawn from a problem distribution* $\mathcal{D}$ *over* $\Pi_{m,p}$, *for any* $h_{RLR}(\lambda, \cdot) \in \mathcal{H}_{RLR}$, *w.p.* $1 - \delta$ *for any* $\delta \in (0, 1)$, *we have*

$$\left| \frac{1}{T} \sum_{i=1}^{T} h_{RLR}(\lambda, P^{(i)}) - \mathbb{E}_{P \sim \mathcal{D}}[h_{RLR}(\lambda, P)] \right| \leq O\left( H\sqrt{\frac{m^2 + \log(1/\epsilon)}{T}} + \epsilon^2 + \sqrt{\frac{1}{T}\log\frac{1}{\delta}} \right).$$

*Proof.* Theorem D.3, 4.3, and Lemma 4.2 directly imply that $\hat{\mathcal{R}}(\mathcal{H}_{\text{RLR}}, \mathcal{S}) = O(H\sqrt{(m^2 + \log(1/\epsilon))/T} + \epsilon^2)$, where $\mathcal{S}$ is a set of $T$ problem instances drawn from problem distribution $\mathcal{D}$ over $\Pi_{m,p}$. Finally, classical uniform convergence bound based on Rademacher complexity concludes the result. $\qquad \square$

### D.4 An extension to 0-1 loss

We now give a formal proof for the learning guarantee of Regularized Logistic Regression hyperparameter tuning problem under 0-1 loss.

**Theorem 4.5 (restated).** *Let* $\tau > 2\epsilon^2$ *and* $\delta \in (0, 1)$, *where* $\epsilon$ *is the approximation step-size. Then for any* $n \geq s(\tau/2, \delta) = \Omega\left( \frac{H^2(m^2 + \log\frac{1}{\epsilon}) + \log\frac{1}{\delta}}{(\tau/2 - \epsilon^2)^2} \right)$, *if we have* $n$ *problem instances* $\{P^{(i)}, \ldots, P^{(n)}\}$ *drawn i.i.d. from some problem distribution* $\mathcal{D}$ *over* $\Pi_{m,p}$ *to learn the regularization parameter* $\lambda^{ERM}$ *for RLR via ERM, then*

$$\mathbb{E}_{P \sim \mathcal{D}}(h_{RLR}^{0\text{-}1}(\lambda^{ERM}, P))) \leq 4 \min_{\lambda \in [\lambda_{\min}, \lambda_{\max}]} \mathbb{E}_{P \sim \mathcal{D}}(h_{RLR})(\lambda, P)) + 4\tau.$$

*Proof.* If $\epsilon < \sqrt{\tau}$, we can rearrange the result in Theorem 4.4 to get

$$s(\tau, \delta) \geq \Omega\left( \frac{H^2(m^2 + \log\frac{1}{\epsilon}) + \log\frac{1}{\delta}}{(\tau - \epsilon^2)^2} \right)$$

samples are sufficient for $(\tau, \delta)-$uniform convergence. Therefore, if $\tau > 2\epsilon^2$, then $\mathcal{H}_{\text{RLR}}$ is PAC-learnable with the ERM algorithm and $s(\tau/2, \delta)$ samples:

$$\mathbb{E}_{P \sim \mathcal{D}}(h_{\text{RLR}}(\lambda^{ERM}, P))) \leq \min_{\lambda \in [\lambda_{\min}, \lambda_{\max}]} \mathbb{E}_{P \sim \mathcal{D}}(h_{\text{RLR}})(\lambda, P)) + \tau.$$

$\qquad \square$