# OpenReview forum: "New Bounds for Hyperparameter Tuning of Regression Problems Across Instances"
_NeurIPS.cc/2023/Conference — NeurIPS 2023 poster_

### Official Review · Reviewer_Ca6P · 2023-06-21

**Soundness:** 3 good
**Presentation:** 2 fair
**Contribution:** 3 good
**Rating:** 7
**Confidence:** 3

**Summary:**

The authors tackle the problem of hyperparameter tuning across problem instances, as proposed by Balcan et al. (2022). They propose three novel learning guarantees regarding the sample complexity of tuning regularization parameters: i) an improved upper bound on the pseudo-dimension for elastic net; ii) a matching lower bound in the same setting; iii) a bound on the pseudo-dimension and on the generalization error for logistic regression.

**Strengths:**

The paper presents new results, which either improve the state-of-the-art (i) or pave a new way (ii and iii) in analyzing common learning techniques. I think these results are topical and interesting for the community.

**Weaknesses:**

As an unschooled researcher regarding the theory of hyperparameter tuning across instances, I found the paper not that easy to read. I particular, I missed the target the authors aimed at (it is stated at the end), which would help me to understand. Here are some remarks which may help in improving the paper:
1) Some elementary definitions are missing and curb the reader understanding (particularly the pseudo-dimension, the Rademacher complexity and the ERM principle in Theorem 4.5). Also, a comment for Theorem 3.1 would be appreciated. As for me, a simple improvement of the paper would be to state informally a kind of Theorem 4.4 at the beginning, such that the reader is more able to understand Sections 3 and 4.
2) Minor comments: “optimization problem 1” should have capital letters, a “be” is missing Line 188, an “of” is missing Line 191, “an GJ algorithm” in the caption of Figure 1 should be “a GJ algorithm”, “Roth et al. ([3])” Line 251 should be “Roth [3]”, increasingness of $R(||\theta||)$ would be better place before stating the representer expansion, the update of $\beta_t^{(\epsilon)}$ in Algorithm 1 should be stored in $\beta_{t+1}^{(\epsilon)}$ instead of $\beta^{(\epsilon)}(\lambda)$, moreover it involves the difference of two vectors of different sizes.

**Questions:**

1) Can the authors define formally the probability space used for the problems (Line 117)? It is a bit disturbing to manipulate objects of varying size.
2) Should it be $h_{b_y}$ and $g^{(1)}(y)$ in Lines 183-184?

**Limitations:**

Limitations are not addressed.

---

> ### Author Rebuttal · Authors · 2023-08-10
>
> We thank the reviewer for constructive feedback and suggestions. We really appreciate that the reviewer acknowledges the theoretical contributions of our paper and we will address the reviewer’s concerns as below. In particular, we will state the target early on in the paper for improving clarity.
>
>
> **Typos in Lines 183-184, and minor comments**: thank you for pointing them out, we will fix those typos in the revised draft.
>
>
> **Additional comments for Theorem 3.1, and 4.4**: thank you for your suggestion. We will add comments and further discussion before the statements of those results in the revised draft for a better presentation.
>
>
> **Adding elementary definitions and results**: unfortunately, due to the space limitation, we could only add some of the elementary definitions in the Appendix (Appendix A). We will try our best to use the additional page in the camera-ready version to integrate the fundamental notions in the main body.
>
>
> **Concerns about varying sizes of problem instances**: nice catch! It is true that this setting allows the number of train ($m$) and validation ($m’$) samples in each problem instance $P$ to vary. It does not even require the feature set across the different problem instances is the same: different problem instances might have different feature sets or even different numbers of features. This makes the settings incredibly general and can handle feature reset, as mentioned in [1].
>
> As mentioned in our work, given a tuple $(m\_i, m’\_i, p\_i)$ of number of train samples, number of validation samples, and number of features, we denote the set of problem instances of that shape $\mathcal{R}\_{m\_i, m’\_i, p\_i}$. Assume that for any problem instance, the validation set can only take at most $m$ samples, and have at most $p$ features, the problem space can be defined as a finite union $\Pi_{m,p} = \cup_{m\_1 \geq 0, m\_2 \leq m, p\_1 \leq p}\mathcal{R}\_{m\_1, m\_2, p\_1}$. Then, the problem distribution $\mathcal{D}$ is simply some unknown distribution over $\Pi_{m, p}$. We will elaborate on this point more carefully in the revised draft.
>
>
> **References:**
>
> [1] Maria-Florina Balcan, Mikhail Khodak, Dravyansh Sharma, Ameet Talwalkar. Provably Tuning Elastic-Net Across Instances. NeurIPS’23

---

> > ### Comment · Reviewer_Ca6P · 2023-08-13
> >
> > I would like to thank the authors for answering my concern regarding the clarity of the paper, and for accepting to improve it.

---

### Official Review · Reviewer_aDmj · 2023-07-06

**Soundness:** 2 fair
**Presentation:** 3 good
**Contribution:** 3 good
**Rating:** 7
**Confidence:** 2

**Summary:**

The paper analyses the complexity of hypotheses classes where the hyperparameter of logistic regression and linear regression are tuned, in the setting where multiple datasets are available.

**Strengths:**

The paper derives tighter upperbounds for the elastic net setting and proves a matching lowerbound, which are extended to KRR, for the setting where regularization parameters are learned from problem instances (e.g. metalearning). The regularized logistic regression setting is tackled by approximating the function class.

**Weaknesses:**

1 While I am quite familiar with learning theory (e.g. VC bounds, PAC, pseudo-dimension) I really have a hard time grasping the problem setting. I had to consult [15] and [27] for that. I think the paper could be greatly improved by clarifying the problem setting in more detail, with some example settings machine learners will be familiar with. E.g. the example of cross validation of [15]. It would also be good to clarify generally what are the inputs to the learner; it is still a bit unclear to me. So are [X_train, y_train, X_test] the inputs? And the output is $\lambda$? The strange thing is, is that the hypothesis seems to take $\lambda$ as input (definition of h), while the $\lambda$ seems the thing we want to learn.

2 Furthermore, in line 124 it is stated "the learning algorithm in this scenario has been mentioned in [15], which is based on simulated annealing" this seems wrong, as the learner in that work [15] is operating in the online learning setting, while the current work seems more to focus on the statistical learning setting. Shouldn't instead be the ERM learner or CV learner of [15] be mentioned?

3 Generally, I think more examples are necessary to grasp the main point of the setting. It would also be clear to further clarify why this is an important problem to study; for which learners can we now derive theoretical guarantees?

4 Once I had clarified the setting I had no time to read the paper furthermore in detail, so I cannot judge the other aspects in more detail.

**Questions:**

see above

**Limitations:**

A challenge is to give a tighter PAC bound for the 01 loss for RLR

---

> ### Author Rebuttal · Authors · 2023-08-10
>
> We thank the reviewer for constructive feedback. We really appreciate that the reviewer acknowledges the contribution of our work and also thank them for their time understanding our work. We will take the reviewer’s comments into account for improving the readability of the paper and use the extra page in the camera-ready version to provide additional illustrations and discussion explaining our setting (see also, attached PDF).
>
>
>
>
> **Improving clarification of the setting**: We will include examples and illustrations to further clarify our setting. For simplicity, consider the LASSO regression problem: the inputs of the learner are problem instances $P$, which are specified by the tuple $P = (X, y, X\_\text{val}, y\_\text{val})$. Given a problem instance $P$ and a regularization parameter $\lambda > 0$, we can define the loss function $h(P; \lambda) = \frac{1}{2} \|\|y\_\text{val} - X\_\text{val} \hat{\beta}\|\|\_2^2$, where $\hat{\beta} = \text{argmin}\_{\beta}\frac{1}{2}\|\|y - X\beta\|\|_2^2+\lambda\|\|\beta\|\|\_1$, and our goal is to learn a good value $\lambda$. So the hypothesis set is the family of regression algorithms $h(\cdot; \lambda)$ parameterized by $\lambda$ (not taking a fixed $\lambda$ as an input), and our goal is to learn a good value $\lambda$ corresponding to a good regularization for typical problem instances $P$. We will emphasize and illustrate this point to clarify the setting in the revised draft.
>
>
>
>
> We agree that including the example of cross-validation in [1] helps clarify the settings of this work and we will include it in the final version of the paper. In particular, $(X_\text{val}, y_\text{val})$ could correspond to validation splits from a fixed dataset to capture usual cross-validation. The setting of course is more general and captures multi-task learning as well where the goal is to learn a common hyperparameter that works well across related tasks.
>
>
> **Comparison with prior work [1]**: While [1] considers both the statistical learning setting and the online setting, we mainly focus on the statistical learning setting improving the results of [1] for the elastic net regression in this setting to obtain asymptotically tight bounds on the pseudo-dimension. We also consider logistic regression which involves technical novelty as no closed-form solution is known.
>
>
> **More discussion on the importance of this line of work**: the problem of tuning regularization parameters in regularized linear models has always been a fundamental issue: choosing inappropriate parameters might cause the model to underfit the data or impair the effect of regularization. In this work, we study a variant of the tuning regularization parameter problem, which involves tuning across multiple problem instances from the same underlying problem distribution. This setting is general and can handle practical scenarios, including the case where the feature set of the problem instances varies or when regression instances from the same domain need to be solved repeatedly. Hence, theoretical analysis for this problem is important and worth exploring.
>
> **Concern about the statement in Line 124**: Thank you for correcting this. It is true that the ERM learner should be mentioned instead.
>
>
> We hope that our answers satisfy the reviewer. If any further clarification is required, please let us know.
>
>
> **References:**
>
> [1] Maria-Florina Balcan, Mikhail Khodak, Dravyansh Sharma, and Ameet Talwalkar, Provably Tuning Elastic-Net Across Instances, NeurIPS’23.

---

### Official Review · Reviewer_QofD · 2023-07-07

**Soundness:** 3 good
**Presentation:** 3 good
**Contribution:** 3 good
**Rating:** 6
**Confidence:** 3

**Summary:**

The main idea of this paper is to address the challenge of tuning regularization coefficients in regression models with provable guarantees across problem instances. The authors investigate the sample complexity of tuning regularization parameters in linear and logistic regressions under l1 and l2 constraints in the data-driven setting. They provide new upper bounds for the pseudo-dimension of the validation loss function class, which significantly improves the existing results on the problem. Additionally, the paper introduces a new approach for studying the learning guarantee in logistic regression regularization coefficients. Overall, the paper aims to contribute to the literature by providing improved bounds and advancements in the field of hyperparameter tuning for regression problems.

**Strengths:**

1. Improved Bounds: One strength of this paper is that it provides improved upper bounds for the pseudo-dimension of the validation loss function class in linear and logistic regressions. These improved bounds contribute to a better understanding of the sample complexity of tuning regularization parameters in regression models.
2. Generality and Applicability: The proposed approach for analyzing the approximation of the original validation loss function class is not limited to regularized logistic regression but can be extended to a wide range of other problems. This demonstrates the generality and applicability of the approach, making it a valuable contribution to the field.


**Weaknesses:**

1. Lack of Experimental Validation: The paper focuses on theoretical analysis and bounds but does not provide experimental validation or empirical results to support the proposed approach. Including experimental validation would have added practical relevance and strengthened the paper's findings by demonstrating the effectiveness of the approach in real-world scenarios.
2. Lack of Clarity and Visual Support: One weakness of this paper is the insufficient clarity in explaining the problem it addresses. The excessive repetition of formulas and the absence of visual aids, such as graphs or charts, make it difficult for readers to understand the main ideas and practical implications of the research. The lengthy theoretical derivations, while comprehensive, may hinder the overall comprehension of the paper's intended message.


**Questions:**

Is it possible to provide any experimental study?

---

> ### Author Rebuttal · Authors · 2023-08-10
>
> We thank the reviewer for providing constructive feedback.
>
> **Clarity and visual support**: We will take your comment into account for the camera-ready version. We already have a figure illustrating the computation of the GJ algorithm (Figure 1) and we are happy to use the additional page in the camera ready version to provide additional illustrations and figures explaining our setting.
>
>
> **Experimental Validation**: Our theory gives bounds on the sample complexity of the ERM and we view our primary contribution as theoretical – including tight bounds improving over previous best-known bounds, and analysis for logistic regression where no closed-form solution is known – we leave the extension of the ideas to practical methods in more application-oriented research for future work.

---

### Official Review · Reviewer_be6e · 2023-07-13

**Soundness:** 3 good
**Presentation:** 3 good
**Contribution:** 3 good
**Rating:** 7
**Confidence:** 2

**Summary:**

This paper studies the sample complexity of tuning regularization parameters in linear and logistic regressions under $\ell_1$ and $\ell_2$ constraints in the data-driven setting. Theoretically, it provides a new bound for the pseudo-dimension of the validation loss function class, which significantly improves the best-known results on the problem. Besides, it provides the matching lower bound. Moreover, it also provides a new bound for tuning the regularization parameters of logistic regression, which previous work cannot do.

**Strengths:**

First of all, I have to admit that I am not an expert in the area of data-driven algorithm design and may miss some related work. Overall, I think this is a solid theoretical work.

- Overall, this paper is well-written and discussions about the comparisons with the related work are clear.
- The considered problem (e.g., hyperparameter tuning or model selection) is important for the machine learning community.
- The theoretical results seem right although I have not carefully checked the proofs line-by-line.
- Theoretically, the obtained upper bound is tight since the (nearly) matched lower bound is provided.
- In contrast with previous work, it is good to provide the analysis for the logistic regression as it might provide techniques to analyze more general settings with non-closed solution forms.

**Weaknesses:**

- Although this is a pure theory work, it would be better to add experimental results to illustrate its applications in practice.
- While the considered setting is the hyperparameter tuning across instances, more discussions can be added to illustrate its practicality over previous work with a single instance.



Minor issues:

typos：
- In lines 503-507 (i.e., Lemma A.1), it is not consistent about the expression of non-increasing (or decreasing) and non-decreasing (or increasing).

**Questions:**

None

**Limitations:**

Please see the weakness part.

---

> ### Author Rebuttal · Authors · 2023-08-10
>
> We thank the reviewer for the thoughtful comments, and for appreciating the theoretical contributions of our work.
>
>
> **Typos in Lines 503-507**: Thank you for pointing out. We will fix the typos in the revised manuscript.
>
>
> **Concerning practicality over single instance work**: In most applications, one needs to solve multiple instances of a regression problem repeatedly, and doing a multi-fold cross-validation on each problem instance can be computationally expensive, Furthermore, collecting all the problem instances to build a single super-instance may not work as examples across instances may not be i.i.d., and moreover may be infeasible to compute as well. We will emphasize this difference in the camera-ready version of the paper.

---

### Author Rebuttal · Authors · 2023-08-10


We thank all the reviewers for their thoughtful comments and suggestions. We are glad that all of the reviewers appreciate our theoretical contribution. The main concerns are the clarity of the settings and more visual supports, which we aim to improve by providing figures describing key concepts of our work. We attached the figures in this rebuttal as a demonstration.


Thank you again for your hard work and feel free to let us know if there is any other concern.

---

### Decision · Program_Chairs · 2023-09-21

**Decision:**

Accept (poster)

**Comment:**

This paper was generally well-received by the reviewers. The results were seen as novel and the problem studied was seen as important. Some reviewers (e.g QofD,aDmj ) raised concerns about the lack of experiments and reviewer (e.gaDmj,Ca6P
 ) raised on the clarity of the presentation and notably on the lack of basic definition and background given on the the mutlti dataset setting considered here. The camera ready version should  correct this and introduce some more formal remarks about the application to cross validation should as the datasets are assumed in section 2 to be sampled i.i.d. In summary a good work that passes the bar for publication at Neurips.